# High time-resolved measurement of stable carbon isotope composition in water-soluble organic aerosols: method optimization and a case study during winter haze in East China

Wenqi Zhang[1,2,3], Yan-Lin Zhang[1,2,3]*, Fang Cao[1,2,3], Yankun Xiang[1,2,3], Yuanyuan Zhang[1,2,3], Mengying Bao[1,2,3], Xiaoyan Liu[1,2,3], Yu-Chi Lin[1,2,3]

1Yale–NUIST Center on Atmospheric Environment, International Joint Laboratory on Climate and Environment Change (ILCEC), Nanjing University of Information Science and Technology, Nanjing 210044, China

2Key Laboratory of Meteorological Disaster, Ministry of Education (KLME)/ Collaborative Innovation Center on Forecast and Evaluation of Meteorological Disasters (CIC-FEMD), Nanjing University of Information Science and Technology, Nanjing 210044, China

3Jiangsu Provincial Key Laboratory of Agricultural Meteorology, College of Applied Meteorology, Nanjing University of Information Science and Technology, Nanjing 210044, China

**Abstract:** Water soluble organic carbon (WSOC) is a significant fraction of organic carbon (OC) in atmospheric aerosols. WSOC is of great interest due to its significant effects on atmospheric chemistry, the Earth's climate and human health. Stable carbon isotope ($\delta^{13}$C) can be used to track the potential sources and investigate atmospheric processes of organic aerosols. However, the previous methods measuring the $\delta^{13}$C values of WSOC in ambient aerosols require large amount of carbon contents as well as time-consuming and labor-intensive preprocessing. In this study, a method of simultaneously measuring the mass concentration and the $\delta^{13}$C values of WSOC from aerosol samples is established by coupling the Gas Bench II preparation device with isotopic ratio mass spectrometry. The precision and accuracy of isotope determination is better than 0.17 ‰ and 0.5 ‰, respectively, for samples containing WSOC larger than 5 μg. This method is then applied for the aerosol samples collected every 3 hours during a severe wintertime haze period in Nanjing, East China. WSOC varies between 3-32 μg m$^{-3}$, whereas $\delta^{13}$C$_{\text{-WSOC}}$ ranges from -26.24 ‰ to -23.35 ‰. Three different episodes (e.g., namely the Episode 1, the Episode 2, the Episode 3) are identified in the sampling period, showing a different tendency of $\delta^{13}$C$_{\text{-WSOC}}$ with the accumulation process of WSOC aerosols. The increases in both the WSOC mass concentrations and the $\delta^{13}$C$_{\text{-WSOC}}$

values in the Episode 1 indicate that WSOC is subject to a substantial photochemical aging during
the air mass transport. In the Episode 2, the decline of the $\delta^{13}C_{\text{-WSOC}}$ is accompanied by the increase
in the WSOC mass concentrations, which is associated with regional-transported biomass burning
emissions. In the Episode 3, heavier isotope ($^{13}C$) is exclusively enriched in total carbon (TC)
compares to WSOC aerosols. This suggests that non-WSOC fraction in total carbon may contain
$^{13}C$-enriched components such as dust carbonate which is supported by the enhanced $Ca^{2+}$
concentrations and air mass trajectories analysis. The present study provides a novel method to
determine the stable carbon isotope composition of WSOC and it offers a great potential to better
understand the source emission, the atmospheric aging and the secondary production of water
soluble organic aerosols.
**Key words:** WSOC, stable carbon, $\delta^{13}C$, aging

**1. Introduction**
Water soluble organic carbon (WSOC) contributes a large fraction (9-75 %) to the organic
carbon (OC) (Anderson, et al., 2008; Decesari et al., 2007; Sullivan et al., 2004) and affects
substantially the global climate change and human health (Myhre, 2009; Ramanathan et al., 2001).
Due to its hydrophilic nature, WSOC has a great impact on the hygroscopic properties of aerosols
and promotes to increase the cloud condensation nuclei (CCN) activity (Asa-Awuku et al., 2011).
WSOC is a contributor to cardiovascular and respiratory problems because it is easy to be
incorporated in biological systems such as human blood and lungs (Mills et al., 2009).
WSOC can be emitted as primary organic carbon (POC) and secondary organic carbon (SOC)
produced from atmospheric oxidation of volatile organic compounds (VOCs) (Sannigrahi et al.,
2006; Weber et al., 2007; Zhang et al., 2018). Due to the hygroscopic property of the WSOC, the
origins of POC may be from biomass burning or marine emissions. However, the SOC may stem
from various sources including coal combustion, vehicle emissions, biogenic emissions, marine
emissions and biomass burning (Kirillova et al., 2010, 2013; Jimenez et al., 2009; Decesari et al.,
2007; Bozzetti et al., 2017b; Bozzetti et al., 2017a).
Stable carbon isotopic composition ($\delta^{13}C$) can provide valuable information to track both
potential sources and atmospheric processes of carbonaceous aerosols (Rudolph, 2007; Pavuluri and
Kawamura, 2012; Kirillova et al., 2013; Kirillova et al., 2014). Carbonaceous aerosols from coal
combustion have an isotope signature from -24.9 ‰ to -21 ‰ (Cao et al., 2011). Particulate matter
emitted from motor vehicles exhibits with isotopes from -26 ‰ to -28 ‰ (Widory, 2006),
respectively. Due to the different pathways of metabolism, C3 and C4 plants exhibit significant
differences of $\delta^{13}C$ (approximately -27 ‰ for C3 and -13 ‰ for C4,   [Martinelli et al., 2002; Sousa
Moura et al., 2008]). Laboratory studies demonstrate that there is no significant isotope fractionation
(± 0.5 ‰) between the produced aerosols and the C3 plants material (Turekian et al., 1998; Currie
et al., 1999; Das et al., 2010). While the C4 plants burning results in $^{13}C$ depletion (< 0.5 to 7.2%)
in the produced aerosols (Turekian et al., 1998; Das et al., 2010). Marine organic aerosol sources
have a carbon isotope signature of -22 ‰ to -18 ‰, (Miyazaki et al., 2011) and play an important
role in the aerosols at coastal sites. In contrast, carbonate carbon exhibits with pretty high isotopic
ratio of -0.3 ‰ (Kawamura et al., 2004), and generally shows a large proportion in dust aerosols.
Thus, the isotope signatures of particulate matter emitted from these various sources may have
different effect on the characteristics of $\delta^{13}C$ in ambient WSOC.
In addition, atmospheric processes like secondary formation and photochemical aging may
change the constitution and properties of WSOC, as well as the stable carbon isotope of WSOC
($\delta^{13}C_{-WSOC}$). According to the kinetic isotope effect (KIE), the reaction rate of molecules containing
heavier isotopes is usually lower than the molecules containing lighter isotopes (Atkinson R., 1986;
Kirillova et al., 2013; Fisseha et al., 2009). The change in reaction rate is primarily results from the
greater energetic need for molecules containing heavier isotopes to reach the transition state (Nina
et al., 1979). Consequently, the oxidants preferentially react with molecules with lighter isotopes
(inverse kinetic isotope effect, KIE), which would result in an enrichment of $^{13}C$ in the residual
materials and a depletion in $^{13}C$ of the particulate oxidation products (Rudolph et al., 2002).
Therefore, organic compounds formed via secondary formation are generally depleted in $^{13}C$
compared with their precursors (Sakugawa and Kaplan, 1995, Fisseha et al., 2009) and this isotope
depletion has proven by   both field measurements and laboratory studies (Pavuluri and Kawamura,
2012). For example, the studies of KIE clearly indicate that the compounds formed via the oxidation
are depleted in the $^{13}C$ compared with their precursors during the reaction of VOCs with OH and
ozone (dominant atmospheric oxidants) (Iannone et al., 2003; Rudolph et al., 2000; Anderson et al.,
2004; Fisseha et al., 2009). Whereas an enrichment of $^{13}C$ in the particulate organic aerosol may
occur in the atmospheric aging processes, such as interactions with photochemical oxidants (e.g.
hydroxyl radical and ozone) during the long range transport. For instance, studies have demonstrated
that the substantial enrichment of $^{13}C$ in the residual, aged aerosols (e.g. isoprene, a precursor of
oxalic acid (Rudolph et al., 2003) after a long range transport. In that case, the stable carbon isotope
can be used to study the sources and the atmospheric processes that contribute to the carbonaceous
aerosols.
Several studies report the temporal and spatial variation, complex chemical species, light
absorption and thermal characteristics of WSOC, as well as its relationship with other compounds
in fine particles (Wozniak et al., 2008; Wang et al., 2006; Zhang et al., 2018; Martinez et al., 2016).
However, only few studies focus on the analysis of $\delta^{13}C$ -WSOC (Fisseha et al., 2006; Kirillova et al.,
2010; Suto et al., 2018; Lang et al, 2012; Zhou et al., 2015). This is partially due to the limited
techniques to analyze the $\delta^{13}C$ signatures of WSOC in ambient aerosols, as their concentrations are
usually very small. In the recent years, some efforts have been made to measure the $\delta^{13}C$ values of
WSOC. Bauer et al. (1991) uses potassium persulfate to convert organic carbon in natural waters
into $CO_2$ for $\delta^{13}C$ measurements. This wet oxidation method requires more than 0.5mM C and 1h
during the pretreatment (from sample injection to the isolation of purified $CO_2$). Fisseha et al. (2006)
boiled the oxidizing solution for 45 min to remove the organic matter and the total time required for
the pretreatment (for 15 samples) is 1.5h. Kirillova et al (2010) develops a combustion method that
applies the aerosol extract without filtration for isotope measurement and involves complicated
processes such as the freeze-drying of the aerosol extract under vacuum for 16 h. This combustion
method is the most widely used for the $\delta^{13}C$ measurements in WSOC aerosols (Kirillova et al., 2010,
2013, 2014; Miyazaki et al., 2012; Pavuluri et al, 2017). Although these methods are able to provide
the $\delta^{13}C$ values of WSOC in natural waters and/or ambient aerosols, the analytical methods require
either large amount of WSOC (from 100 μg C to 0.5 mM C) or time-consuming preprocessing. And
some of the methods oxidize the WSOC extract without filtration and/or decarbonation in the
pretreatment., which would result in higher uncertainty of the $\delta^{13}C$ results. The high detection limit
of the previous methods is difficult to determine the $\delta^{13}C_{-WSOC}$ in aerosol samples with low carbon
concentrations. In that case, an easily operated method detecting the $\delta^{13}C_{-WSOC}$ values in aerosol
samples with low detection limit and high precision is urgently needed. The objectives of this study
are: 1) to provide an accurate, precise and easily operated method to measure the WSOC and $\delta^{13}C_{-}$
$_{WSOC}$ in ambient aerosol samples. 2) to apply this method for analyzing the high time resolution
aerosol samples during a severe haze and discuss the potential sources and the atmospheric
processes of WSOC. In addition, the concentrations of inorganic ions and air mass back trajectories
coupled with MODIS fire maps are also analyzed to substantiate the results obtained from the $\delta^{13}C$
analysis.
**2. Methods**
2.1 Standards
Four working standards are used in this study: potassium hydrogen phthalate (KHP), benzoic
acid (BA), sucrose ($CH_6$) and sodium oxalate ($C_2$). KHP and BA are widely used as the standards
of WSOC measurements (Kirillova et al., 2010) and then are used here as the WSOC test substances.
Also, their isotope signatures are close to the $\delta^{13}C$ values of aerosol samples (Miyazaki et al., 2012;
Fisseha et al., 2009; Suto et al., 2018). Sucrose and oxalic are taken as the standards to represent the
characteristics of the components in atmospheric WSOC (Fowler et al., 2018; Liang et al., 2015;
Pathak et al., 2011; Pavuluri and Kawamura, 2012). The carbon isotope composition of these four
standards are: -12.20 ‰ ($CH_6$), -13.84 ‰ ($C_2$), -27.17 ‰ (BA) and -30. 40 ‰ (KHP), respectively.
The wide range of the delta values of the working standards is able to cover the majority of the $\delta^{13}C_{-}$
$_{WSOC}$ values in ambient aerosol samples. Standards are resolved in Milli-Q water (18.2 MΩ quality)
to make standard solutions of 0.25μg mL$^{-1}$, 0.75 μg mL$^{-1}$, 1.5μg mL$^{-1}$, 3μg mL$^{-1}$, 6μg mL$^{-1}$, 12μg
mL$^{-1}$ and 24μg mL$^{-1}$, which means containing carbon content of 1ug, 3ug, 6ug, 12ug, 24ug, 48ug
and 96ug in 4mL standard solution to test the procedures during the pretreatment.
2.2 Aerosol samples
The aerosol samples are collected during a severe haze in January (from Jan 14[th] to 28[th]) of
2015 at the suburban of Nanjing, a megacity in East China. The sampling site is located at the
Agrometeorological station in the campus of the Nanjing University of Information Science and
Technology. It is close to a busy traffic road and surrounded by a large number of industrial factories.
PM$_{2.5}$ samples are collected on pre-combusted quartz-fiber filters (180×230mm) every 3 hours with
a high-volume aerosol sampler (KC100, Qingdao, China) at a flow rate of 1 m$^3$ min$^{-1}$. After
sampling, all the filters are wrapped in the aluminum foil, sealed in air-tight polyethylene bags and
stored at -26 ˚C for later analysis. A field blank is obtained by placing the blank filter in the filter
holder for 10 minutes without sampling.
2.3 Chemical analysis
PM$_{2.5}$ concentrations are observed at Pukoku Environmental Supervising Station.
Concentrations of total carbon (TC) and $\delta^{13}C_{-TC}$ values are analyzed with EA-IRMS (Thermo Fisher
Scientific, Bremen, Germany). WSOC mass concentrations are measured with the TOC analyzer
(Shimadzu). Ion concentrations are obtained from Ion Chromatograph (IC, Thermo Fisher Scientific,
Bremen, Germany). Besides, the meteorological data are observed nearby the sampling site (Enivs
automatic meteorological station).
2.4 Sample pretreatment
The wet oxidation method is used to covert the WSOC to $CO_2$ (Sharp J. H., 1973), and the
resulting $CO_2$ can be measured by IRMS. The overview of the optimized method for measuring
WSOC and $\delta^{13}C_{-WSOC}$ in the aerosols is shown in Fig. 1. The process of the pretreatment consists of
6 steps: WSOC on a 20 mm diameter disc is extracted with 6 mL mili-Q water through water-bath
ultrasonic for 30 minutes (step 1-2). The WSOC extract is filtered with a 0.22 μm syringe filter to
remove the particles in step 3. 2.0 g potassium persulfate (K$_2$S$_2$O$_{8,}$ Aladdin Industrial Corporation,
Shanghai) and 100 μL phosphoric acid (85 % H$_3$PO$_4$, AR, ANPEL Laboratory Technologies Inc.,
Shanghai) are dissolved in 50 mL Milli-Q water to make the oxidizing solution. The oxidizing
solution made within 24 h is added into the filtered WSOC extract as shown in step 4 of Fig. 1. The
phosphoric acid is added to remove the inorganic carbon resolved in the solution, and the persulfate
is added for the preparation to convert the organic compounds to $CO_2$. The vails are sealed tightly
with the caps as soon as the oxidizing solution is added into the WSOC extract.
To remove the ambient $CO_2$ dissolved in the mixture (mixed solution of the oxidizing solution
and the WSOC extract) and the atmospheric $CO_2$ in the headspace of the sealed sample vials, high-
purity helium (Grade 5.0, 99.999 % purity) is flushed into the vials for 5 min in step 5. The aim of
this step is to exclude the possible contamination from the atmospheric $CO_2$, and it has to be finished
within 12 hours after the mixture of the WSOC extract and the oxidizing solution to avoid the loss
of $CO_2$ produced under room temperature. High-purity helium (15-18 mL min$^{-1}$) is flushed under
the water surface and a stainless steel tube is set for the output gas stream. The open end of this tube
is submerged in Milli-Q water to prevent any backflow of atmospheric $CO_2$ (Fig. 1., step 5). After
flushing, the vials are heated at 100 ˚C for 60 min in the sand bath pot (quartz sand, Y-2, Guoyu,
China) to start the oxidation of WSOC in step 6. The heated vials are stored overnight at room
temperature for condensing the moisture before the analysis on IRMS to prevent the damage to the
measuring equipment.
2.5 Determination of the carbon content and stable carbon isotopic ratios

$CO_2$ gas produced in the headspace of the prepared sample is extracted and purified by Gas

Bench II (Gas Bench II, Thermo Fisher Scientific, Bremen, Germany), and introduced into an
isotope ratio mass spectrometer (IRMS) (Mat 253, Thermo Fisher Scientific, Bremen, Germany)
for $\delta^{13}C_{-CO2}$ analysis. The extracted gas is purified with a Nafion water trap to remove the water
vapor and then the gas is loaded into a 100 uL sample loop through an eight-port Valco valve. After
120 s loading time (the duration time from the beginning of the analysis to the first rotation of the
eight port in the Gas Bench II.), the eight-port Valco valve rotates every 70 s to inject the sample
gas from the loop into a GC column (Poraplot Q fused-silica cap, 25 m, 0.32 mm; Agilent
Technologies). The GC column is set at 40 ˚C for the $CO_2$ separation from the matrix gases. The
separated $CO_2$ is introduced into another Nafion water trap and subsequently enters into the IRMS
with an open split. The $CO_2$ gas in each vial is detected 10 times in 15 minutes, showing 10 sample
peaks after five reference peaks. The peak areas and the isotope compositions of the 10 sample
peaks are given correspondingly, the results of the first two sample peaks are abandoned considering
the possible memory effect of the system. The average peak area and the isotope composition of the
last eight peaks is taken as the result of a certain sample determined by GB-IRMS.
**3.  Method optimization**

The wet oxidation method is adapted from the stable isotope analysis of organic matter in

ground water (Lang et al., 2012; Zhou et al., 2015). Several tests are performed to adjust the optimal
conditions for measuring WSOC aerosols with relative low carbon amounts.
3.1 The carbon content in the procedural blank

In order to quantify the low concentration of WSOC in aerosols, it is critical to reduce the

carbon content in the procedural blank for minimizing the detection limit of the method. To achieve
this goal, the procedural blanks are analyzed to test the contamination that the reagents would
introduce to the results (shown in Table 1). The average carbon content in the procedural blank is
about 0.5 μg C (corresponding with a peak area of 0.23 Vs) with a $\delta^{13}C$ value of -27.04 ± 1.28 ‰
(n=15). The carbon contents and the isotope compositions of Mili-Q water and the agents dissolved
in the oxidizing solution are also determined to identify the source of contamination in the
procedural blank. The peak area of Mili-Q water is not detected (Table. 1.) after going through all
the processes in the pretreatment without adding any other materials, suggesting no contamination
is introduced from the Mili-Q water. After that, the contamination from 85% $H_3PO_4$ with different
purity (acid-1: analytical reagent, AR; acid-2: High Performance Liquid Chromatography, HPLC)
are compared. The carbon contents in the 85% $H_3PO_4$ dissolved in Mili-Q water are 0.03-0.04 μg C
and show no significant discrepancy between different purity.

Interestingly, the carbon content increase to 0.5-0.6 μg C after the persulfate is added,

implicating that the $CO_2$ in the procedural blank is mainly produced from the oxidation of organic
substance in the persulfate. The carbon content in HPLC grade of 85% $H_3PO_4$ mixed with the
persulfate (0.58 - 0.63 μg C) is closed to that of AR grade (0.46 - 0.63 μg C, see table 1.). Thus, AR
grade with purity of 85 % $H_3PO_4$ is utilized to prepare the oxidizing solution in this method. The
average carbon content of the procedural blank is estimated to be 0.5 ±0.06 μg C, and the detection
limit is expected to be 10 times the procedural blank (i.e. 5 μg C). The carbon content in the
procedural blank of this method is much lower than that of the methods analyzing isotopes of WSOC
in aquatic environment or soil (De Groot, 2004; Polissar et al., 2009; Werner et al., 1999). The
smaller carbon content of the procedural blank suggests the possibility to correctly measure the
WSOC and $\delta^{13}C_{-WSOC}$ of samples containing low carbon content.
3.2 Flushing methods

To avoid any contamination, the headspace of the sample vial has to be flushed with the high-

purity helium to remove the $CO_2$ (both dissolved and gas phase). Two different flushing methods
(F1 and F2) are compared here. F1 is a one-step flushing: helium is bubbled under the water surface
for 5 min in a sealed vial, and the gas in the headspace is released through a stainless steel tube to
the atmosphere. The open end of this tube is submerged in Milli-Q water to balance the air pressure
and to prevent any backflow of the atmospheric $CO_2$. F2 requires two steps: the helium is first
bubbled under the water surface for 5 min in an open vial to remove the dissolved $CO_2$ in the solution.
After the vial is sealed, the helium is flushed again into the headspace for 5 min by piercing the
septum with a two-hole sample needle. The two holes are performed as the inlet of the helium and
the exit of the outflow, respectively. Since the flow rate of the inlet helium is larger than that of the
outflow, the headspace pressure is considered to be greater than 1atm. In that case, the most
noticeable difference between F1 and F2 is the air pressure of the headspace.
Different concentrations of working standard (KHP) are tested to compare the flushing
methods. The results obtained from F1 and F2 show no significant difference regardless of the
concentration of KHP. This represents that F1 and F2 are both able to completely remove the $CO_2$
in the vials. But it has to be noticed that F2 produces excessive air pressure in the headspace, the
following heating step may increase the risk of gas leak. Gas leaking during the preparation usually
results in the loss of carbon content and the isotope fractionation. Besides, flushing with F2 takes 5
more minutes for each sample compared with F1. Consequently, F1 is considered as the suitable
flushing method to remove $CO_2$ dissolved in the solution and the headspace.
3.3 Heating time
In order to assure the complete oxidation of WSOC, duration time for heating the samples is
tested with KHP, a widely used WSOC standard which is difficult to oxidize. Figure 2 shows the
carbon contents and the $\delta^{13}C$ values of KHP solutions heated from 15 min to 120 min at 100℃.
Some caps of the sample vials are out of shape after heating for longer time (more than 60 min),
suggests gas leak of the vials. High pressure can be built up in the headspace with the increase of
the temperature during the long time heating, especially for the vials containing more carbon
contents. The $CO_2$ gas produced in the headspace may leak through the minor holes on the septum
pierced by the stainless tube during the helium flushing step (step 5 in Fig 1.). According to the
kinetic isotope effect (KIE), isotope fractionation occurs during the gas leaking. The light carbon
isotopes ($^{12}C$) are easier to escape from the vials than the heavy ones ($^{13}C$), thus the remaining $CO_2$
would be more enriched with heavy isotopes ($^{13}C$). In that case, lower carbon contents and higher
$\delta^{13}C$ values are expected to be observed in the results of leaking vials. In the results of the KHP
standards, some of the vials containing larger amount of organic carbon are detected to have
extremely low carbon contents corresponding with very high isotopic ratios. For example, one of
the 10 μg C KHP standard is measured to be 1.2 μg C and $\delta^{13}C = 14.9$ ‰ after 120 min of heating;
one of the 30 μg C KHP standard is measured to be 2.4 μg C and $\delta^{13}C = 17.7$ ‰ after 90 min of
heating (Fig. 2.). The stable results (both carbon contents and the isotopes, Fig. 2.) of 4 μg C
standards are probably due to the less $CO_2$ gas and lower pressure produced in the headspace during
the heating. Accordingly, heating time longer than 60 min increases the probability of gas leak in
the measurement.
In the aspect of the isotope composition, KHP standards heated for 15min, 30min and 60 min
all show stable results with similar standard deviations (from 0.51 - 0.57, see Table S1). While, the
heating time of 15min and 30 min are not long enough for the complete oxidation, which is shown
in lower carbon contents (Fig. 2.). Therefore, heating for 60 min at 100℃ is found to be the most
suitable to produce constant results without gas leak and isotope fractionation.
3.4 Waiting time and instrument settings
The waiting time of the mixture (the aerosol extract and the oxidizing solution) between step 4
and 5 in Fig. 1. is tested to prevent the $CO_2$ loss during the flushing. Some of the compounds in
aerosol samples could be oxidized at room temperature. The $CO_2$ generated from the mixture before
heating could be lost during the flushing step (Sharp, 1973). The ambient sample is tested to detect
the room - temperature - oxidized $CO_2$ (Fig. S1.). Replicates of the ambient aerosol extract (from
one filter) are mixed with the oxidizing solution, and the mixtures of the aerosol extract and the
oxidizing solution are flushed with He to exclude the effect of $CO_2$ (both in the headspace and in
the mixture) as soon as possible. After flushing, the mixtures are stored at room temperature from
1 to 31 hours before analysis without heating. The carbon contents produced in the mixtures that
stored less than 12 h before analysis is smaller than 0.02 μg, which contributes to ~ 7% to the carbon
content in the procedural blank (0.5μg C). But when the waiting time is extended to 31 h, up to 2.3
μg C (about 5 times of the procedural blank) is oxidized into $CO_2$. The room - temperature - oxidized
$CO_2$ produced during the waiting time would be flushed out by the He in the later procedure and
then would result in significant isotope fractionation in the delta results. Therefore, the mixture
should be flushed with He within 12 h to avoid the $CO_2$ loss and isotope fractionation.
In addition, various combinations of shorter loading times (30-90 s) and/or fewer sample peaks
(i.e. 5 sample peaks) are tested with reference gas ($CO_2$ mixed with He) to shorten the analysis in
the system. However, the amount of $CO_2$ in the reference gas detected by the mass spectrometry is
about 2 μg C lower compared the results obtained with longer loading times and more sample peaks.
And there is a decrease of isotope value (~ 0.4 ‰) as well when the loading time is shorter or the
sample peaks are less than 10. Thus, 120 s loading time and 10 sample peaks are necessary for the
precise results, and the standard deviation is < 0.03 ‰ for the 10 sample peaks within a run.
3.5 Calibration of the results
3.5.1 Quantification of the carbon content
The sample peak area is proportional to the carbon content in the vial and then is used to
quantify the amount of $CO_2$ in the inflow of IRMS. The average value of the peak areas for the last
eight sample peaks is taken as the peak area of a certain sample. The first two sample peaks are
excluded to avoid the effect of the residual $CO_2$ of the former vial. We established a carbon content
standard curve (linear equation) by measuring the peak areas of $CO_2$ gas samples containing 1-24
μg C (Fig. S2.). It has to be noted that the gas samples containing larger carbon contents are not
tested for the difficulty of injecting too much volume of $CO_2$/He gas. Then the amount of $CO_2$
oxidized from the unknown samples can be quantified with this linear equation (i.e., Carbon content
(μg) = Peak area (Vs) × (2.50 ± 0.08) − (0.62 ± 0.39), $R^2$=0.98). The standard curve (linear equation)
of the peak areas against the carbon contents in the WSOC solution (KHP solution containing 1-100
μg C) is also established (Fig. S2.). And a linear equation similar with the peak areas against $CO_2$
gas is obtained (i.e., Carbon content (μg) = Peak area (Vs) × (2.34± 0.01) − (0.86± 0.14), $R^2$=1.00).
Then the conversion efficiency of the WSOC extract containing 1-100 μg C can be roughly
calculated as 104 ± 3 %. The high conversion efficiency demonstrates the completely conversion
and the negligible isotope fractionation during the oxidation. In that case, the carbon content in the
WSOC extract of unknown samples can be calculated based on the standard curve of peak areas
against the carbon content in the WSOC extract. And the standard curve quantifying the carbon
content has to be established with every batch of unknown samples to assure the completely
conversion.
3.5.2 Blank correction
The blank contribution to the WSOC mass concentrations and the $\delta^{13}C_{-WSOC}$ values are
evaluated with the peak area and the isotope value of the procedural blank. The peak area (average
value of the last eight peaks) from the measurement is proportional to the carbon content in the vial
and then is taken to represent the $CO_2$ amounts in the inflow of IRMS. The procedural blank can be
corrected according to the mass balance as follows.
$$\delta^{13}C_{meas} \times A_{meas} = \delta^{13}C_{corr} \times (A_{meas} - A_{blk}) + \delta^{13}C_{blk} \times A_{blk} \qquad (1)$$
Where $\delta^{13}C_{corr}$, $\delta^{13}C_{meas}$ and $\delta^{13}C_{blk}$ are the blank-corrected $\delta^{13}C$, the measured $\delta^{13}C$ of the samples
and the $\delta^{13}C$ of the procedural blank, respectively. $A_{meas}$ and $A_{blk}$ denote the peak areas of the
samples and the blank, correspondingly.
In order to calibrate the contribution of the procedural blank to the isotope results, $A_{blk}$ and
$\delta^{13}C_{blk}$ are calculated with an indirect method (Polissar et al., 2009). KHP ($\delta^{13}C$= -30. 40 ‰) and
$CH_6$ ($\delta^{13}C$= -12.20 ‰) with various concentrations are measured to calculate $A_{blk}$ and $\delta^{13}C_{blk}$. The
wide range of their isotopes can basically cover the $\delta^{13}C_{-WSOC}$ values in most ambient aerosol
samples. According to Eq. (1), $\delta^{13}C_{meas}$ can be written as the following:
$$\delta^{13}C_{meas} = \delta^{13}C_{corr} + A_{blk}(\delta^{13}C_{blk} - \delta^{13}C_{corr})/A_{meas} \qquad (2)$$
According to Eq. (2), there is a linear relationship of the $\delta^{13}C_{meas}$ values and the reciprocal of
peak areas ($1/A_{meas}$). Based on the keeling plot theory, linear equations of the $\delta^{13}C_{meas}$ values and
$1/A_{meas}$ for the two standards can be set up separately (e.g., $\delta^{13}C$ and $1/A_{meas}$ values obtained from
the measurement of $CH_6$ and their linear relationship are shown in Fig.S3.). The slopes (k1 and k2)
and the intercepts (b1 and b2) of this liner relationship can be expressed with $\delta^{13}C_{blk}$, $A_{blk}$ and $\delta^{13}C_{corr}$
as follows.
$$\begin{aligned} k_1 &= A_{blk} \times (\delta^{13}C_{blk} - \delta^{13}C_{corr-std1}) \\ k_2 &= A_{blk} \times (\delta^{13}C_{blk} - \delta^{13}C_{corr-std2}) \end{aligned} \qquad (3)$$
$$\begin{aligned} b_1 &= \delta^{13}C_{corr-std1} \\ b_2 &= \delta^{13}C_{corr-std2} \end{aligned} \qquad (4)$$
Thus, $A_b$ and $\delta^{13}C_b$ can be calculated as follows:
$$\delta^{13}C_{blk} = (k_2 \times b_1 - k_1 \times b_2)/(k_2 - k_1) \qquad (5)$$
$$A_{blk} = (k_2 - k_1)/(b_1 - b_2) \qquad (6)$$
Thus, the blank contribution is able to be calibrated with the equation below:
$$\delta^{13}C_{corr} = (\delta^{13}C_{meas} \times A_{meas} - \delta^{13}C_{blk} \times A_{blk})/(A_{meas} - A_{blk}) \qquad (7)$$
For example, $\delta^{13}C_{blk}$ and $A_{blk}$ are calculated to be -27.43‰ and 0.3Vs (~0.5 µg C) based on the
results of KHP and $CH_6$ (shown in Fig. 3.). The carbon content in the procedural blank contributes
to $1 - 10\%$ carbon content of an ambient aerosol sample. Although the $\delta^{13}C_{blk}$ and $A_{blk}$ are not
strongly varied values, they need to be measured before every batch of the ambient samples to assure
the stable status of the system (IRMS) and the proper processes during the pretreatment.
3.5.3 Calibration of isotope results
In order to calibrate the isotope results, four working standards (KHP, BA, CH$_6$ and C$_2$)
containing different carbon contents are measured with EA-IRMS and Gas Bench II-IRMS. The
standards measured with EA are combusted at 1000℃ to convert the organic materials into $CO_2$ for
the measurement in IRMS without pretreatment. More than 10 repetitions of each standard are
measured in this way, the average delta values (after blank correction) of each standard are defined
as correct values here. On the other hand, the average isotope compositions (after blank correction)
of 10 repetitions obtained from the wet oxidation method (determined with Gas Bench II) are
defined as measured values. Thus the calibration curve can be established on the basis of the
measured values and the correct values (Fig. S4.). For instance, the isotope results can be calibrated
as follows:
$$\delta^{13}C_{cali} = k \times \delta^{13}C_{blk-corr} + b \tag{8}$$
$\delta^{13}C_{cali}$ is the isotope composition after the isotope calibration, $\delta^{13}C_{blk-corr}$ is the blank corrected
isotope composition determined with Gas Bench II, k and b are the slope and the intercept obtained
from the calibration curve. Similar with the blank correction, the isotope calibration curve needs to
be established with each batch of the ambient samples to assure the stable status of the IRMS and
the proper processes during the pretreatment.
In this way, the isotope results can be calibrated, the raw data and the isotope composition after
the blank correction and the isotope calibration determined with Gas Bench II are compared in Fig.
3. The correct values of standard carbon isotopes are plotted in Fig. 3. as well. The isotope results
after two steps of correction (the blank correction and the calibration of isotope results) are closer
to the correct values (isotopes measured with EA) and the blank contribution are drastically
eliminated. But as for the standards containing carbon content smaller than 5 μg C, the contribution
of the procedural blank (with an isotope ratio about -27.43‰) is still significant. According to the
isotope variation of the ambient aerosols, the analysis of isotope compositions is not reliable if the
repetitions of the standards show difference larger than 1‰ (SD > 0.5 ‰). After correction, the
standard deviations of isotope results of each standard are better than 0.17 ‰ (regardless of the
carbon content of a certain standard) when the carbon contents are larger than 5 μg C. In that case,
the detection limit of this method is 5 μg C and the results (both carbon contents and the isotopic
ratios) of WSOC lower than 5 μg C were not reliable.)
3.6 QA/QC procedure

A batch of working standards with different carbon contents are measured to evaluate the

optimized method in this study (data shown in Fig. 3.). The quality of the unknown samples is
assured with a standard curve established with the peak areas and the corresponding input carbon
contents of WSOC extract (e.g. in Fig. S2.). The conversion efficiency of the WSOC oxidation is
104 $\pm$3% . The average recovery of the working standards and the ambient samples are tested to be
97 $\pm$ 6 % and 99 $\pm$ 10 %, respectively. The conversion efficiency and the recoveries suggest
completely oxidation of WSOC extract without significant isotope fractionation in the pretreatment.
The blank contribution is evaluated with the peak area and the isotopic ratio, these values are
calculated with the indirect method introduced in Sect. 3.5.2. According to the carbon content (0.3
- 0.5 $\mu$g C) and the isotope composition ($\sim$ -27.43 ‰) of the procedural blank, the WSOC detection
limit of this method is 5 $\mu$g C, 10 times of the carbon content in the procedural blank. The blank
corrected isotope compositions should be calibrated again with the calibration curve as described in
Sect. 3.5.3 to obtain the isotopic ratios of the unknown samples.
In order to obtain the carbon contents and the corrected isotope compositions of the unknown
samples, at least two kinds of standards need to be measured before every batch of the unknown
samples. The range of the carbon contents and the isotope compositions of the standards are required
to cover the range of WSOC and $\delta^{13}C_{-WSOC}$ in the ambient samples, e.g. KHP, BA and $CH_6$. Hence,
the concentration standard curve, the linear equations for the blank correction and the isotope
calibration curve are able to be established according to the results of the standards. Besides, one
standard should be measured after every 10 unknown samples to assure the stable status of the
equipment.
As for the isotope measurement, the precision of the last eight sample peaks is < 0.15 ‰ within
a run for standards containing more than 1 $\mu$g C; between runs, the deviation of the standards with
different carbon contents (> 5 $\mu$g C, n $\geq$ 10) is < 0.17 ‰. The accuracy is estimated to be better than
0.5 ‰ by comparing the calibrated $\delta^{13}C$ results from Gas Bench II and the blank corrected isotopic
ratios from EA. Isotope results tested by Gas Bench II is slightly lower compared to the results of
EA. The ambient aerosol filters are tested repeatedly to evaluate the reproducibility of the ambient
samples as well. The standard deviation of the WSOC concentrations and the isotope results of the
repeated ambient samples are 0.25 $\pm$ 0.04 $\mu$g C (n$\geq$3) and 0.14 $\pm$ 0.07 ‰ (n$\geq$3), respectively. To
conclude, the presented method is considered to be precise and accurate to detect the low abundance
of WSOC as well as isotopes in aerosol samples.
To test the applicability of this method measuring the atmospheric WSOC, the ambient aerosol
samples collected in Nanjing are analyzed. And the WSOC concentrations are measured with TOC
analyzer (Shimadzu) for comparison. Figure 4. shows the scattered plot of WSOC concentrations
measured with the two peripherals (TOC analyzer and Gas Bench II-IRMS). The strong correlation
($R^2 = 0.95$, p<0.01) and the slope (0.97) demonstrate the reliability of measuring WSOC with the
presented method. It suggests complete oxidation of WSOC in aerosol samples, which means no
significant carbon isotope fractionation happens during the preparation. Moreover, the $\delta^{13}C_{-WSOC}$
values (between -26.24 ‰ to -23.35 ‰) of ambient aerosols are close to the published data (from -
26.5 ‰ to -17.5 ‰) (Kirillova et al., 2013; Kirillova et al., 2014). In that case, the $\delta^{13}C$ values
resulted from this method are considered to be effective for ambient WSOC.
**4. Sources and atmospheric processes of WSOC**
4.1 Temporal variation
Time series of $PM_{2.5}$, $\delta^{13}C$ values, chemical tracers and meteorological data observed at the
sampling site during the studied period are illustrated in Fig. 5. WSOC ranges from 3.0 to 32.0 μg
$m^{-3}$, occupying $49 \pm 10$ % of total carbon in $PM_{2.5}$. The stable carbon isotopes of WSOC and TC
vary between -26.24 ‰ to -23.35 ‰ and -26.83 ‰ to -22.25 ‰, respectively. $\delta^{13}C$ values shift over
2 ‰ in 24 hours, and over 1 ‰ in 3 hours, which is not able to be captured in lower time resolution
samples (e.g., 12h or 24h). In that case, this data set can be interpreted with more detailed
information about the WSOC sources and the atmospheric processes. Biomass burning tracer (nss-
$K^+$), dust tracer ($Ca^{2+}$), MODIS fire spots and air mass trajectories are analyzed to investigate the
potential sources of WSOC. Nss-$K^+$ is used as a proxy of biomass burning (Zhang et al., 2013). Nss-
$K^+$ concentrations are evaluated from $Na^+$ concentrations in the samples according to their
respective ratios ($K^+/Na^+$=0.037 *w/w*) in seawater (Osada et. al., 2008).
$$nss - K^+ = [K^+] - 0.037 \cdot [Na^+] \tag{9}$$
where $[K^+]$ and $[Na^+]$ are the total mass concentrations of $K^+$ and $Na^+$ of the aerosol samples.
The concentration of nss-$K^+$ ranges from 0.16 to 6.70 μg $m^{-3}$ with an average of 1.31 μg $m^{-3}$. The
high concentrations and the intense increase in Jan 24th indicate a significant biomass burning event
and will be discussed later.
As shown in Fig. 5., $\delta^{13}C_{-TC}$ and $\delta^{13}C_{-WSOC}$ show similar pattern during the sampling period. In
general, $\delta^{13}C_{-TC}$ is slightly lower than $\delta^{13}C_{-WSOC}$, and the trend is also observed elsewhere (Fisseha
et al., 2009). The difference is related to the sources and the atmospheric processes during the
formation and transformation of carbonaceous particles in the atmosphere. The C4 plants biomass
burning and the marine organic materials are the sources with relatively enriched $^{13}C$. Smith and
Epstein (1971) suggest that C4 plants have a mean $\delta^{13}C$ isotope signature of -13 ‰. And the isotope
composition of carbon emitted from phytoplankton, an example of primary marine aerosol, is about
-22 ‰ to -18 ‰ (Miyazaki et al., 2011). However, January is not a specific time period for the
growing or combustion of C4 plants in East China, indicating small possibility of C4 plants biomass
burning as a major source of WSOC aerosols. In addition, both WSOC and non-WSOC components
can be emitted from biomass burning, thus the C4 plants combustion would generally result in the
enrichment of $^{13}C$ in both TC and WSOC. The air parcel transported from marine areas normally
has little effect on the aerosols during winter in Nanjing (Qin et al., 2016), suggesting the negligible
contribution of marine emissions to WSOC during the sampling period. Therefore, the WSOC
sources with higher isotope signatures (compared with non-WSOC sources) are not able to explain
the higher values of $\delta^{13}C_{-WSOC}$ over $\delta^{13}C_{-TC}$.
Apart from the sources, the secondary formation (Hecobian et al., 2010; Jimenez et al., 2009;
Saarikoski et al., 2008) of WSOC is reported to affect the isotope compositions. Precursors like
VOCs can be oxidized with the hydroxyl radicals and ozone to produce WSOC in the atmosphere
(Pathak et al., 2011). Laboratory and field studies demonstrate that the lighter isotopes have the
priority to be oxidized and produce particulates with lower isotopic ratios. For example, the
oxidation of VOCs in the atmosphere would result in the $^{13}C$ depletion in the products and the $^{13}C$
enrichment in the residual VOCs (Rudolph et al., 2002). In other words, the secondary formation
tends to lower the $\delta^{13}C$ value of ambient WSOC, thus the secondary formation could not explain
the $^{13}C$ enrichment in WSOC compared to TC.
Studies demonstrate that the photochemical aging process during the long range transport
causes significant enrichment in $^{13}C$. (Aggarwal and Kawamura, 2008; G. Wang et al., 2010). The
isotope fractionation is up to 3 ‰ - 7 ‰ of the residual during the photolysis of oxalic acid, a
dominant species in WSOC aerosols (Pathak et al., 2011). Due to the hydrophilic property, WSOC
is associated with the aerosol aging processes. WSOC/OC ratio is normally considered to represent
the aging status of aerosol samples (Agarwal et al., 2010; Pathak et al., 2011), it increases with the
photochemical aging process. The ratio of WSOC/OC is 0.67 $\pm$0.12 (Fig. S5.) in this study, which
is higher than the aged aerosols with WSOC/OC = 0.41 reported elsewhere (Huang et al., 2012).
The high ratio of WSOC/OC indicates aged aerosols during the sampling period. Thus the
photochemical aging process could partially explain the reason of higher values of $\delta^{13}C_{-WSOC}$
(compared with $\delta^{13}C_{-TC}$).

According to the principle of mass balance, $^{13}C$ depleted sources of non-WSOC can also result

in the depletion of $^{13}C$ in TC. TC is consist of OC, EC and carbonate carbon (CC) (Huang et al.,
2006), and OC can be divided into WSOC and water insoluble OC (WIOC) according to the
hydrophilic character (Eq. 10). In most circumstances, CC is negligible to the amount of TC in $PM_{2.5}$
(Huang et al., 2006; Ten Brink et al., 2004), thus non-WSOC component could be presented as Eq.

11.

$$TC = OC + EC + CC = WSOC + WIOC + EC + CC \qquad (10)$$

$$TC - WSOC = WIOC + EC \qquad (11)$$

WIOC and EC are generally originated from primary emissions (Park et al., 2013; Y. L. Zhang

et al., 2014), and the $\delta^{13}C$ values are better representing their sources. In that case, the $^{13}C$ depleted
source which only contributes to non-WSOC components, such as WIOC emitted from the
vegetation, is likely to be another reason of $\delta^{13}C_{-TC}$ depletion during the sampling period.
4.2 Three episodes

During the sampling period, three significant haze events (e.g., namely the Episode 1, the

Episode 2, the Episode 3) are observed in Nanjing. These 3 episodes show different tendencies of
$\delta^{13}C_{-WSOC}$ variation during the accumulation of WSOC aerosols (see Fig. 5.). The Episode 1 and 2
are compared here due to the distinct $\delta^{13}C_{-WSOC}$ trends with WSOC accumulation. In the Episode 3,
$^{13}C$ is found to be enriched in TC compared to WSOC ($\delta^{13}C_{-WSOC} < \delta^{13}C_{-TC}$, p<0.01), in contrast to
the trend of isotope compositions during other periods ($\delta^{13}C_{-WSOC} > \delta^{13}C_{-TC}$, p<0.01).
4.2.1 The Episode 1

As for the Episode 1, the $\delta^{13}C_{-WSOC}$ values increase with the mass concentrations of WSOC (r

= 0.84, p < 0.001, see Fig. 6d.), indicating the sampling site is impacted by $^{13}C$ enriched WSOC
sources and/or photochemical aged aerosols. As shown in Fig. 6a., air mass trajectories of WSOC
with higher $\delta^{13}C_{-WSOC}$ values (>24‰) are originated mainly from northern China, and the northerly
wind prevails at this site (Fig. 5g.). During the long-range transport, the studied WSOC mass
concentration increases with the $^{13}$C enrichment of WSOC due to the isotope fractionation in the
photochemical aging process. This is supported by the increasing ratio of WSOC/OC (from 0.73 to
0.91) in the Episode 1 (Fig. S5.).

According to the higher isotopes ($\delta^{13}$C-WSOC > -24 ‰) and the corresponding trajectories (Fig.

6a.), C4 plants biomass burning ($\delta^{13}$C ~ -12‰, [Martinelli et al., 2002; Sousa Moura et al., 2008])
and coal combustion ($\delta^{13}$C ~ - 24.9 ‰ to -21 ‰, [Cao et al., 2011]) are considered to be possible
sources of WSOC. Nss-K$^+$ is largely originated from plants combustion (Zhang et al., 2013), and is
analyzed as a proxy of biomass burning. However, during this period the nss-K$^+$ level (0.56 $\pm$0.41
μg m$^{-3}$) is not significantly increased and is generally lower than the average value (1.3 μg m$^{-3}$,
Fig.5e), indicating that the C4 plants biomass burning is not a major source of WSOC. Besides, the
main crops growing in northern China are mainly C3 plants such as wheat and rice instead of C4
plants during the sampling period (Chen et al., 2004). And the biomass burning contribution of C3
plants would even lower the $\delta^{13}$C values of WSOC. What's more, there are only few MODIS fire
spots along with the trajectories from northern China (Fig. 6a.). In that case, open field biomass
burning is not considered as a major source of WSOC at the sampling site during the Episode 1.

Furthermore, the WSOC mass concentrations and the $\delta^{13}$C-WSOC values decrease synchronously

with the change of the wind direction (from north to southeast) after the Episode 1. The southeast
wind breaks the continuous transport of WSOC from northern China. And the relatively lower $\delta^{13}$C-
WSOC values are then observed, suggesting a regional isotope signal of WSOC without the substantial
aging. Besides, the WSOC/OC declines obviously with the isotope after the Episode 1 (Fig. S5.),
indicating less contribution of aged aerosols to the sampling site. Therefore, the elevated $\delta^{13}$C-WSOC
values with the increased WSOC mass concentrations in the Episode 1 are mainly affected by the
aged aerosols transported from northern China.
4.2.2 The Episode 2

The $\delta^{13}$C-WSOC values show an opposite trend with WSOC mass concentrations (r = -0.54, p <

0.01, see Fig. 6e.) in the Episode 2. At the beginning of the Episode 2 (Jan 22$^{nd}$), the sampling site
is mainly affected by the air mass from the north of Nanjing, and the WSOC displays with relatively
higher $\delta^{13}$C-WSOC values at the same time (Fig. 6b). After Jan 22$^{nd}$, the shift of the wind direction
and the air mass trajectories are well corresponded with the decline of the $\delta^{13}$C-WSOC values (Fig.
6b.). The large amount of fire spots in the potential source regions suggests the significant impact
of open field biomass burning. It should be noted that the stable carbon isotope composition of C3
plants combustion is relatively low (i.e., $\delta^{13}C \sim$ -27‰, [Martinelli et al., 2002; Sousa Moura et al.,
2008]). The $\delta^{13}C_{-WSOC}$ values decrease and the WSOC mass concentrations peak to the maximum
when the air mass travels throughout the regions with a great many hot spots. The concentration of
nss-$K^+$ has a positive correlation with WSOC concentration (r = 0.82, p < 0.001) and a negative
correlation with $\delta^{13}C_{-WSOC}$ (r = -0.45, p < 0.05) during the Episode 2. And the concentration of nss-
$K^+$ increases up to 6.7 μg m$^{-3}$, about 7 times of the average value, indicating a significant biomass
burning contribution (Fig. 5e). The decrease of the $\delta^{13}C_{-WSOC}$ values and the increase of the biomass
burning tracers (i.e., nss-$K^+$) suggest that the biomass burning emission is a major contribution of
WSOC aerosols. Also, the WSOC/OC ratio declines from 0.88 to 0.53 (Fig. S5.), indicating that the
increased WSOC is rather from fresh biomass-burning aerosols without a substantial aging process.
4.2.3 The Episode 3
The $^{13}C$ is clearly enriched (p<0.01) in TC (-23.5 ± 0.43 ‰) compared to WSOC (-25.17 ±
1.08 ‰) during the Episode 3 (see Fig. 6f.). This might be related with a $^{13}C$-enriched source and/or
the aging process of non-WSOC fraction in TC. Non-WSOC fraction is mainly consist of WIOC,
EC and carbonate carbon (CC). Among these carbonaceous species, carbonate carbon (CC) exhibits
with much higher $\delta^{13}C$ values than EC and OC (Kawamura et al., 2004). CC could be a significant
fraction of dust aerosols, even though it is a very small part of TC in PM$_{2.5}$ in most cases.
To study the dust contribution in the Episode 3, $Ca^{2+}$ is determined as an indicator of dust
(Huang et al., 2010; Jankowski et al., 2008). $Ca^{2+}$ and TC show similar patterns ($R^2 = 0.84$, p < 0.01),
indicating dust origins in this period. The argument is also supported by the 48-h backward
trajectory analysis (Fig. 6c.). It shows that the air mass is mainly originated from a semi-arid region,
Mongolia. The photochemical aging of dust aerosols during the long-range transport from Mongolia
to Nanjing could possibly promotes the $^{13}C$ enrichment. For short, the enrichment of $^{13}C$ in TC over
WSOC is due to a dust event transported to the studied site.
According to the mass balance, the isotopic ratio of TC affected by CC in the dust aerosols can
be expressed as follows:
$$\delta^{13}C_{-TC} = f_{cc} \times \delta^{13}C_{-CC} + (1 - f_{cc}) \times \delta^{13}C_{-NC} \qquad (12)$$

where the $\delta^{13}C_{-TC}$, $\delta^{13}C_{-CC}$ and $\delta^{13}C_{-NC}$ are the measured stable carbon isotope of TC, the isotopic

ratio of CC in dust aerosols and the isotope composition of non-CC fractions. The $f_{cc}$ represents the

CC contribution to TC. The CC contribution during the Episode 3 is roughly estimated based on a

few assumptions: 1) the increase of TC and $\delta^{13}C_{-TC}$ is only affected by the dust origin, 2) the average

value of $\delta^{13}C_{-TC}$ (-25 ‰) during the studied period (except the Episode 3) is taken as the value of

$\delta^{13}C_{-NC}$, 3) $\delta^{13}C_{-CC}$ = 0.3 ‰ in dust sources (Kawamura et al., 2004). With these considerations, CC

contribution is estimated to contribute up to 10% to TC according to the Eq. 12.

**5. Conclusions**

An optimized method for the determination of WSOC mass concentrations and $\delta^{13}C_{-WSOC}$

values in aerosol samples with Gas Bench II - IRMS is presented. A two-step correction is applied

to correct the blank contribution and to calibrate the isotope results. The procedural blank is

estimated to be 0.5 μg C with isotope composition of -27.43 ‰. The detection limit is demonstrated

to be 5 μg C according to the measurement of working standards with various carbon contents. The

method yields a high recovery of the standards (97 ± 6 %) and ambient samples (99 ± 10 %).

According to the high recoveries, the isotope fractionation during the pretreatment is tend to be

negligible. The precision and the accuracy is better than 0.17 ‰ and 0.5 ‰, separately. WSOC

concentrations determined with this optimized method is consistent ($R^2$ = 0.95) with the results of

the TOC analyzer. Compared with the previous methods, the optimized method presented in this

study is more precise and accurate, and requires less time-consuming pretreatment.

The presented method is then applied to analyze the $\delta^{13}C_{-WSOC}$ of the high time resolution

aerosol samples collected during a severe winter haze in East China. WSOC ranged from 3.0 μg m$^-$

$^3$ to 32.0 μg m$^{-3}$, and $\delta^{13}C_{-WSOC}$ varies between -26.24 ‰ to -23.35 ‰. $^{13}C$ is more enriched in WSOC

than TC in the majority of the sampling period, indicating aged aerosols and/or $^{13}C$ depleted primary

sources of non-WSOC component. Three haze events (e.g., namely the Episode 1, the Episode 2,

the Episode 3) are identified with different tendencies of $\delta^{13}C_{-WSOC}$ during the accumulation of

WSOC aerosols. Similar patterns of the WSOC concentrations and the $\delta^{13}C_{-WSOC}$ values in the

Episode 1 are demonstrated to be affected by the air mass transported from northern China. The

increase of $\delta^{13}C_{-WSOC}$ indicates that the WSOC aerosols from the studied site is subject to a

substantial photochemical aging process during the long range transport. The contrasting trend of

the WSOC and $\delta^{13}C_{-WSOC}$ values in the Episode 2 is interpreted as the contribution of regional C3

plants biomass burning sources. In the Episode 3, the heavier isotope ($^{13}$C) is clearly enriched in
total carbon (TC) compares to WSOC fraction due to the dust contribution.

The optimized method is demonstrated to be accurate and precise to detect the WSOC mass

concentration and its isotope compositions ($\delta^{13}$C$_{-WSOC}$) in aerosols. Our results indicate that the high
time-resolved measurement of $\delta^{13}$C$_{-WSOC}$ can be used to distinguish different atmospheric processes
such as photochemical aging and aerosol sources (e.g., biomass burning and dust). However, a
quantitative understanding of sources and formation processes of WSOC aerosols is still of great
challenge. To reduce the knowledge gaps, a combination of multiple methodologies is needed in
future studies, such as high time-resolved measurement of radiocarbon ($^{14}$C) and stable carbon
isotope compositions ($\delta^{13}$C), and the real-time measurement of chemical compositions (e.g.,
Aerosol Mass Spectrometers, AMS or Thermal Desorption Aerosol Gas Chromatograh-AMS).

*Author contributions.* YZ conceived and designed the study; YZ, FC and WZ designed the
experimental strategy; WZ and YX performed the sampling and isotope measurements; YZ and WZ
analyzed the experimental data; YZ and WZ proposed the hypotheses; WZ wrote manuscript with
YL; all other co-authors contributed to writing.

*Competing interests.* The authors declare that they have no competing interests.

*Acknowledgements.* This study is supported by the National Natural Science Foundation of China
(Grant nos. 91644103, 41761144056, 41603104), the National Key Research and Development
Program of China (Grant no. 2017YFC0212704) and the Provincial Natural Science Foundation of
Jiangsu (Grant no. BK20180040).

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

**Table 1.** Various blank preparation with results.

| Identifier | Oxidant [a] | Acid [b] | C content ($\mu$gC) | $\delta^{13}C$(‰) |
|---|---|---|---|---|
| Mili-Q water | - | - | ND* | - |
| Mili-Q water | - | - | ND* | - |
| Mili-Q water +Acid-1 | - | 100 uL 85 % $H_3PO_4$, AR | 0.04 | -1.6 |
| Mili-Q water +Acid-1 | - | 100 uL 85 % $H_3PO_4$, AR | 0.04 | -4.3 |
| Mili-Q water +Acid-2 | - | 100 uL 85 % $H_3PO_4$, HPLC | 0.03 | -4.9 |
| Mili-Q water +OX+Acid-1 | 2.0 g $K_2S_2O_8$ | 100 uL 85 % $H_3PO_4$, AR | 0. 63 | -25.90 |
| Mili-Q water +OX+Acid-1 | 2.0 g $K_2S_2O_8$ | 100 uL 85 % $H_3PO_4$, AR | 0.54 | -25.69 |
| Mili-Q water +OX+Acid-1 | 2.0 g $K_2S_2O_8$ | 100 uL 85 % $H_3PO_4$, AR | 0.46 | -24.77 |
| Mili-Q water +OX+Acid-2 | 2.0 g $K_2S_2O_8$ | 100 uL 85 % $H_3PO_4$, HPLC | 0.63 | -26.66 |
| Mili-Q water +OX+Acid-2 | 2.0 g $K_2S_2O_8$ | 100 uL 85 % $H_3PO_4$, HPLC | 0.56 | -27.38 |
| Mili-Q water +OX+Acid-2 | 2.0 g $K_2S_2O_8$ | 100 uL 85 % $H_3PO_4$, HPLC | 0.58 | -26.91 |

[a, b] oxidant and acid are added to 50 mL Mili-Q water.
ND* : Not detected

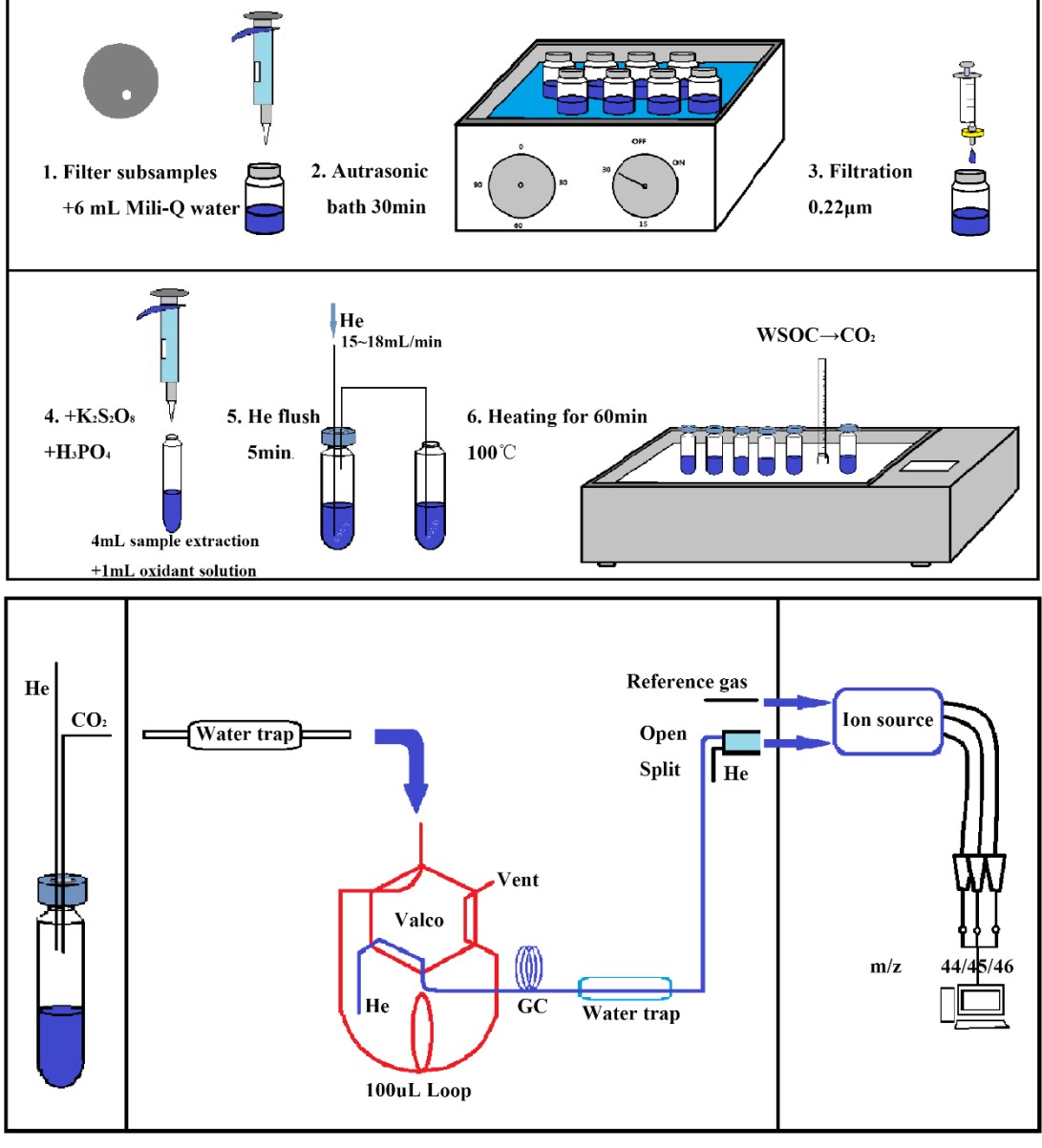


**Figure 1.** Schematic of the optimized method for the measurement of WSOC mass concentrations and the $\delta^{13}C$ $_{-WSOC}$ values. (A filter disc is dissolved with 6mL Mili-Q water in a 20 mL pre-combusted glass bottle in the first step. After 30 minutes autrasonic bath, the WSOC extract is filtered with 0.22 μm syringe filter and transferred to another 20 mL pre-combusted glass bottle in step 3. 4 mL filtrate is transferred to a 12 mL pre-combusted glass vial which contains 1 mL oxidant solution (2.0g $K_2S_2O_8$ and 100 μL 85% $H_3PO_4$ dissolved in 50 mL Mili-Q water) in the vial in step 4. Next, the mixed solution of WSOC extract and the oxidant solution is flushed with Helium at a flow rate of 15-18 mL min$^{-1}$ as shown in step 5. At last, the vials are heated for 60 minutes under 100 ℃ in the sand bath pot (step 6).)

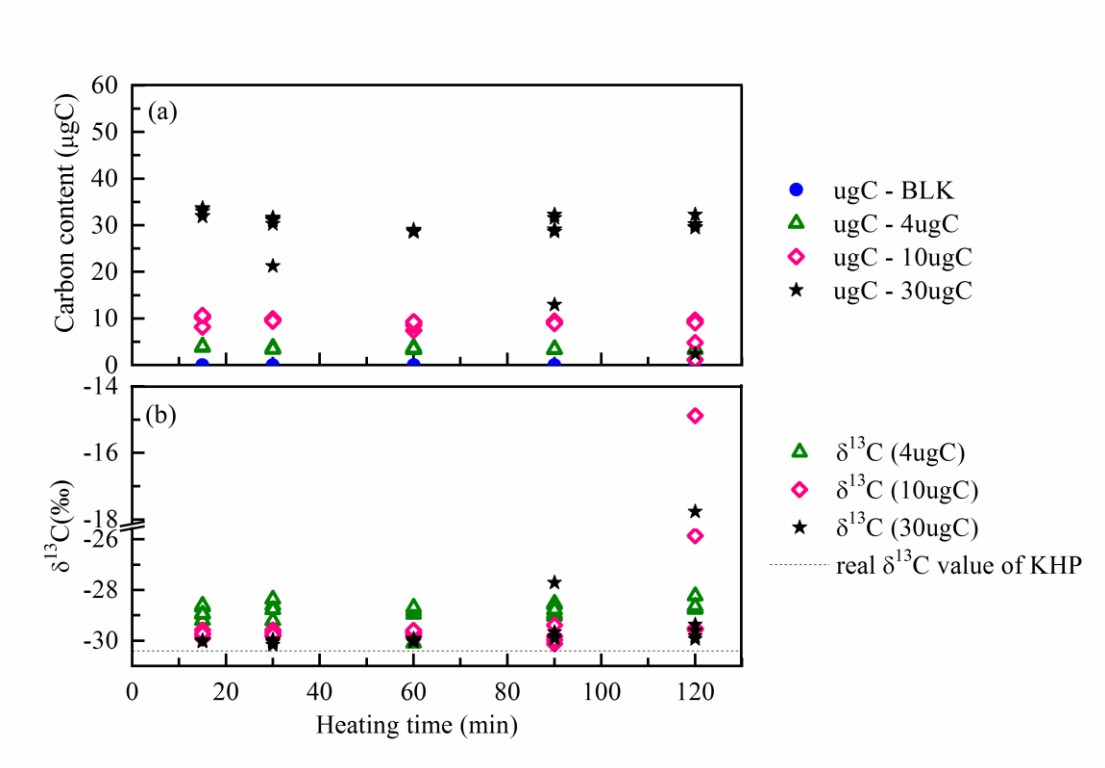


**Figure 2.** Carbon contents (a) and isotopic ratios (b) of KHP after different heating time.

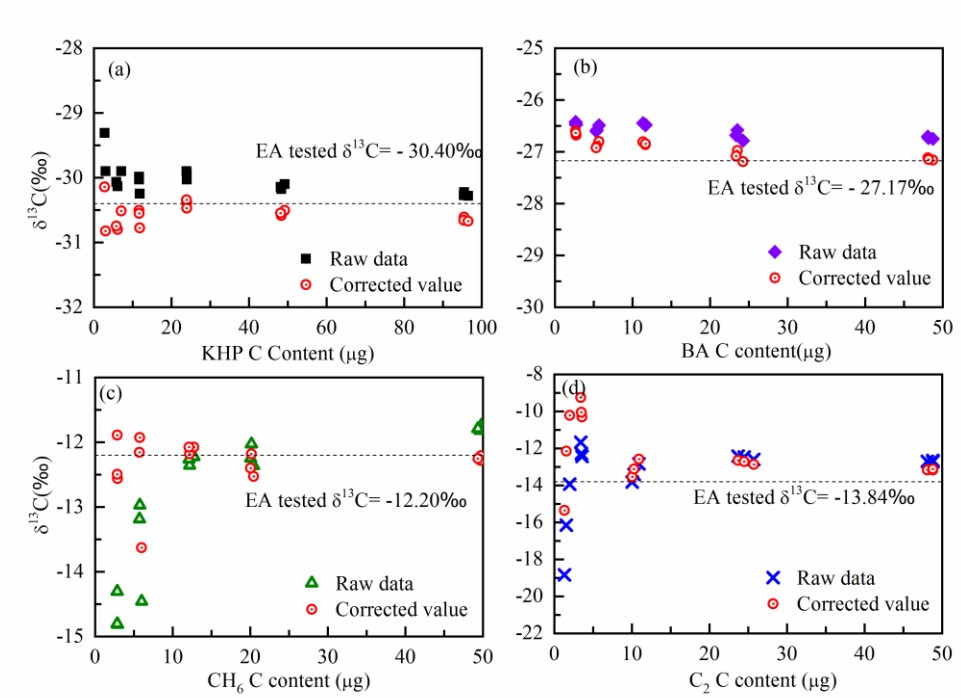


**Figure 3.** Isotope results before and after the two-step correction of the four standards.
(a. KHP, b. BA, c. CH$_6$, d. C$_2$. Red circle with a spot represents the two-step corrected isotopic
ratios; ■,◆,△,× represent the raw data from Gas Bench II; the dotted line represents the blank
corrected δ$^{13}$C values tested by EA)

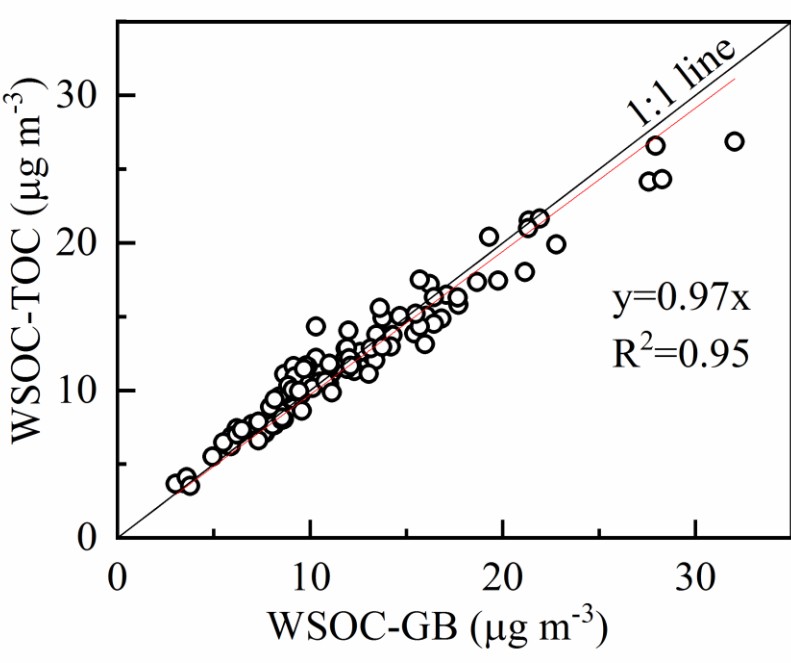


**Figure 4.** Correlation of WSOC mass concentrations measured with Gas Bench II - IRMS and
TOC analyzer.


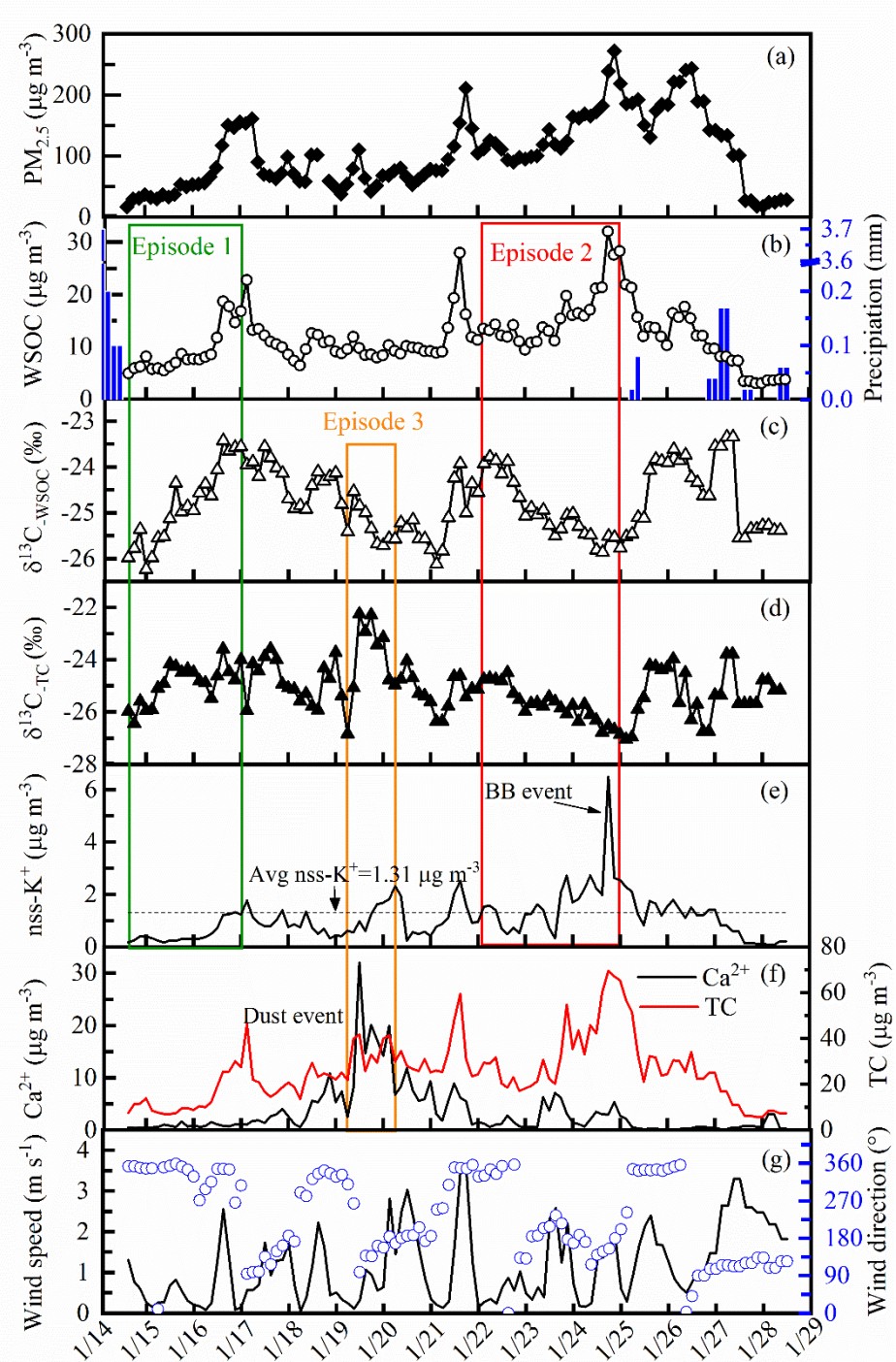

**Figure 5.** Time series of PM$_{2.5}$, WSOC, precipitation, $\delta^{13}$C values, nss-K$^+$, Ca$^{2+}$, TC, wind speed and wind direction at the sampling site during the studied period. (The time period framed with the rectangles is defined as the Episode 1 (green), the Episode 2 (red) and the Episode 3 (orange). The dotted line in 5e is the average value of nss-K$^+$ during the studied period. The high concentration and intense increase of nss-K$^+$ in the Episode 2 indicate a significant biomass burning (BB) event, and is marked with "BB event" in 5e. The similar trends of Ca$^{2+}$ and TC suggest a dust event in the Episode 3.**)**

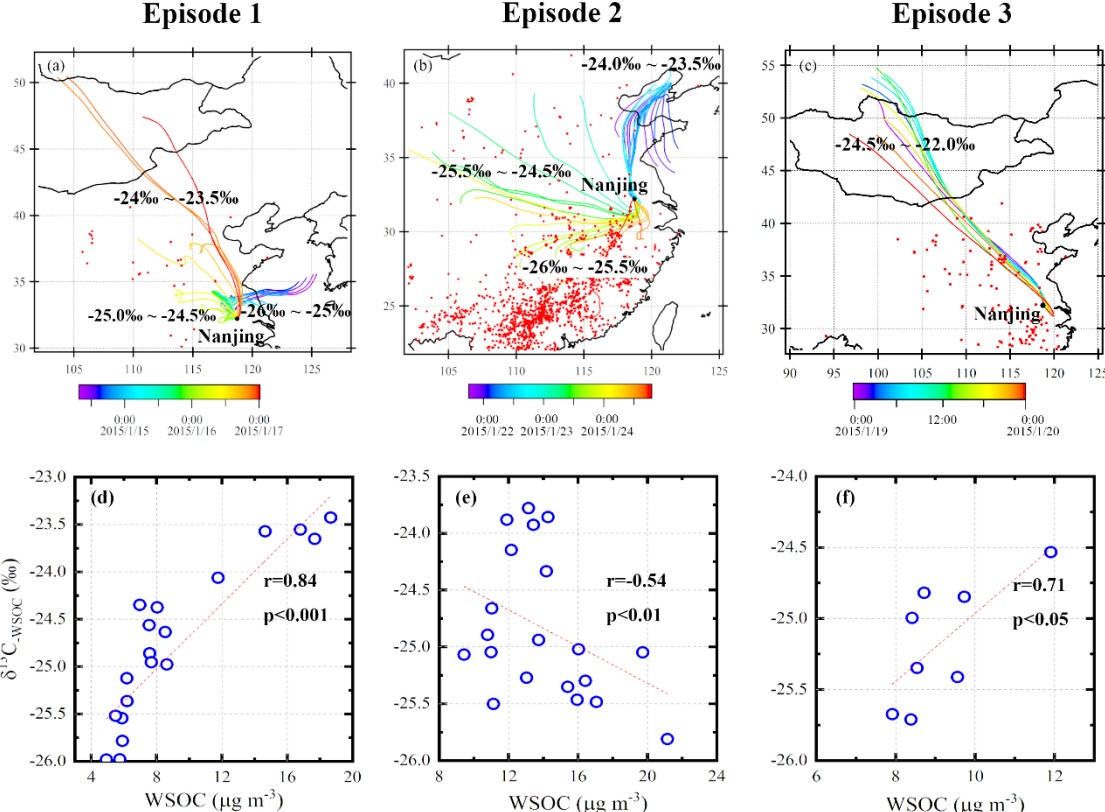

936

**Figure 6.** 48h-air mass back trajectories at 500m and MODIS fire maps in the three episodes and

the corresponding relationship between WSOC and $\delta^{13}C_{-WSOC}$.

(a, b and c represent the back trajectories and the fire maps of the Episode 1, 2 and 3, separately.

The colors of the back trajectories are marked according to the time of the specific trajectory. Red

points represent the fire spots in each episode obtained from the Fire Information for Resource

Management System (FIRMS) derived from the Moderate Resolution Imaging Spectroradiometer.

The ranges of the $\delta^{13}C$ values of the back trajectories are labeled: the marked isotopic ratios are the

$\delta^{13}C_{-WSOC}$ values (for a and b) and the $\delta^{13}C_{-TC}$ values (for c). d, e and f are the correlation between

WSOC and $\delta^{13}C_{-WSOC}$ in each episode.)