# Peer review of "High time-resolved measurement of stable carbon isotope composition in water-soluble organic aerosols: method optimization and a case study during winter haze in East China"

_Atmospheric Chemistry and Physics, 2018_

## Referee Comment (RC1) · Anonymous Referee #2 · 2 Mar 2019

General comments:
* * *
The manuscript shows the setup and thorough evaluation of a method to analyze 13C in small amounts of water-soluble organic carbon (WSOC). This method is then applied in an interesting case study and demonstrates the high time resolution that can be achieved by analyzing such small sample amounts. Measuring 13C accurately and precisely on aerosol samples as small as 5 $\mu$g is challenging and it is nice to see that the authors succeed in this. My recommendation is major revisions, because

(1) part of the system testing and of the analysis procedure needs to be explained

much more clearly and in more detail, before it is really understandable (see specific comments), ie. Probably 1-2 additional figures are required for the supporting material

(2) All the reproducibility tests were done with standard only, but the reproducibility of actual filter samples is usually lower and this should be tested as well. Please select a couple filters (with high and low loading) and analyze them repeatedly to get a better idea of reproducibility for WSOC extracted from filter samples

(3) The writing clarity and the grammar need to be improved, see specific comments for examples, but try to proofread the whole manuscript

Specific comments:

————————————————-

Line 47/48: In don't understand what this sentence means and it seems out of context. What is meant by "surface layer" and does the transport refer to vertical transport (e.g. deposition) or some other transport, such as long-range transport? Unless you explain in much more detail, omit this sentence

Line 61: Please provide reference with respect to fractionation during biomass burning

Line 69-78: Please explain this in a bit more detail and avoid unclear formulations, e.g. "lighter isotopes (12C) have the priority to be oxidized", eg mean that "molecules containing lighter isotopes usually react faster than molecules containing heavier isotopes" (and this is still an oversimplification, but more acceptable than what is written here; "isotope depleted matters": The matter is not depleted in isotopes, it is depleted in the heavy isotope (13C), "secondary formed WSOC" = WSOC formed by secondary processes, "positive isotopic fractionation" = I think you mean enrichment here? Line 74: what does "these" refer to.

Please reformulate this entire paragraph and use as precise and correct language as possible

Line 89: This sentence seems out of context here, please explain this in more detail in a separate paragraph

Paragraph 2.2: The description is not clear. The figure itself quite clear and informative, but the text is not. The figure contains several steps, it would help to make clear in the text, which step (e.g., 1, 2,3 . . .) you are talking about. I think for clarity, you should refer to the extract from the filters as "WSOC extract" instead of extraction. After you add the oxidizing solution you name the resulting mixture sometimes as "mixture", sometimes as "solution", this is confusing – choose one. Make clear that the removal of $CO_2$ from the mixture refers to dissolved ambient $CO_2$ (i.e. a cleaning step before oxidation). People, who don't know the subject well might assume that just adding the oxidizing solution would lead to oxidation. Finally make clear (in the text) that heating in the sand bath results in the oxidation of WSOC to $CO_2$.

Line 126: What gases are typically in the headspace after oxidation?

Line 159: This sentence might fit better in the method section

Line 164: Here you speak about "oxidizing agents" in plural, but in the text only one ($H_3PO_4$), is discussed. What about the other components of the solution?

Line 167: "blank effect of $H_3PO_4$" unclear formulation – I think you mean that you determine the carbon content of $H_3PO_4$, which will introduce contamination (blank) in your analysis.

Line 170-172: To make this more clear state the procedural blank (average +-std)before and after addition of $H_3PO_4$

Line 190: Mention again here again that KHP is a standard. It was introduced in the very beginning, but most people will not remember the abbreviation several pages later

Line 203: "overheated" sounds like too high temperature, not too long heating times

Paragraph 3.3/Figure 2: Looking at Figure 2, I cannot find the trends and effects de-

scribed in the text. I think a statistical test would show no significant differences be-
tween concentration or d13C values at different heating times. I can also not detect
that data are less scattered at 60min. Vs (10ug) is maybe more scattered at 120 min
than for other heating times, but at the same time the isotope values at 120 min are
much more stable. Vs - 30ug has some big outliers for 120, 90 and 30min, but no
corresponding outliers in d13C. On the other hand d13C (4ug) has outliers at 60 and
30 min, when Vs-4ug is particularly stable. Overall I see data point with comparable
scatter and occasional outliers, I think the text over-interprets the results.

Line 221: Confusing formulation – the carbon content of the WSOC samples stays the
same. Only the contamination detected during flushing is lower for samples stored less
than 12 hrs . . .

Line 225: purged? Do you mean "heated and analyzed"?

Line 226/227: "tested through reference gas detection" What do you mean by this?

Line 227/228: The sensitivity of a mass spectrometer does not depend on sample
peaks and loading time. What do you mean by "sensitivity of mass spectrometry"?

Line 228/229: what do you mean by "decline in isotopic ratios"? A decrease in delta
values (i.e. depletion)? Or a decrease in reproducibility? Or something else?

Section 3.5.1: This section describes the method for blank correction, but barely any
results. Please also give the results at least in the supporting material

Line 248 – 251: Show a graph of d13C_meas vs 1/A in the appendix, what was the
typical R2?

Line 255-256: What were the values you obtained for d13C_blk and A_blk? How does
A_blk compare to A of a typical sample?

Section 3.5.2: This section is very superficially described and very difficult to under-
stand, partially due to poor English. I'm not 100% sure what the authors are doing

here. It definitely needs to be carefully rewritten with more precise language and more detail. But if I am not completely mistaken by the vague descriptions, this section just describes a calibration, where delta values are normalized against standard material that is treated the same way as the samples?

Line 259: I don't think "system errors" is the correct term to describe what you are investigating here, "systematic bias" is probably a better term.

Line 261/262: The logic here is reversed: Systematic errors are the cause of differences between measurements on different systems, not the other way around.

Line 264: I assume larger amounts of the standard material analyzed on the EA (presumably without pretreatment) are taken as the "correct value" against which the standard materials that have undergone the whole extraction procedure are evaluated. If this is correct, then please describe clearly.

Line 266: what do you mean by "isotope standard curve", a calibration curve? Please show an example, at least in the appendix.

Line 267: Which "two different peripherals"? This whole sentence does not make sense to me.

Line 269: I don't see raw values from EA in the figure, does the EA result not give the nominal value (horizontal line)? "Corrected results . . ." In figure 3 the corrected data are labeled as: "blank corrected" but the text suggests that they are both blank corrected and normalized (calibrated). Which one is true?

Line 272: What do you mean by "dilution curves". Dilution curves do not have a precision . . .

Section 3.6 This section contains a summary of the previous test results. What this section should contain is a description of how the quality of the unknown samples are assured. How is the blank correction done exactly, i.e. which values are taken for d13C_blank and A_blank? How many and which working standards are measured

with the samples, and how frequently i.e. was the calibration curve described above only established once, or is it measured every day? Etc . . . please re-write this section.

Line 349: At least for PM2.5, not for PM10

Line 355: What could such a source be?

Line 376: primary OC from coal combustion does not contain much WSOC

Minor comments:
* * *
Line 84: "Previous method*s*". Please have the manuscript proofread for similar grammar errors throughout

Line 162: "optimistic" – do you mean "optimal"?

Line 234: replace "blank effects of the" with "blank contribution to"

Line 235: " ..could represent" -> is proportional to

Line 411: "Exclusively" -> clearly

---

## Referee Comment (RC2) · Anonymous Referee #1 · 9 Mar 2019

The manuscript reports an optimized method to analyze the 13C in the WSOC with small amounts, then applied it in the high time resolution aerosol samples during the haze period, and try to explain the source and processes beyond the variations of $\delta$13C-WSOC. Although the context of this study does not introduce a bran-new method to analyze the carbon isotope or the new atmospheric process beyond what has been known, the optimized method will be a valuable contribution to the knowledge of WSOC. Overall, the manuscript is organized and read well. The paper does have a number of major issues that need to be addressed before publication.

Major comments:

[Figure]

(1) The manuscript set up an optimized method, and declare it is novel, but limited information for previous method (e.g. the theory, operation, use condition and deficiency…) present in the introduction section. Thus, the reader cannot get the challenge or difficulty for the establish of new method. Try to add the previous researches in the introduction, the challenge of the new method in the abstract, and the comparison in the discussion section.

(2) Line 55-65: The isotope signatures of particulate matter (not the WSOC, or VOCs) emitted from different sources have limited the various of isotope signatures of $\delta$13C-WSOC, due to the most part of the WSOC come from the secondary formation rather than the primary emission. In addition, the isotopic effects during the formation or aging process is absence of proof, need add some reference.

(3) The method needs to be explained much more clearly and in more detail, including the flow rate of He, the volume of the bottle (6 mL extraction for the 20 mm diameter disc, but how much for the TOC analysis, and how much use for the $\delta$13C-WSOC analysis, if all use for $\delta$13C-WSOC analysis, and another one for TOC, then the results in fig 4 need be reconsider due to the maldistribution in one filter), the volume of sample extracted solution, the ratio of sample extracted solution to the oxidizing agents….

(4) adjust the 2.4 and 2.5 section after 2.1 section

(5) The conversion efficiency (removal efficiency) showed be test, which is important to quality assurance of the $\delta$13C-WSOC results. That is because there is fractionation during the conversion processes if the conversion efficiency is low, especially during a long reaction time. The operations of removal efficiency should cover two aspects: the residual WSOC in the reaction solution and the amount of $CO_2$ in the bottle after heating, and compare them with the added WSOC.

(6) The IRMS method for WSOC concentration showed be present detail in the method section, rather than a single sentence like "Line 236 the peak area obtained from the measurement could represent the carbon content". If it means using the peak area

represent the CO2 amounts in the inflow of IRMS? If yes, then show which peak use (10 peak total?), the gas volume in bottle and into IRMS (in the atm pressure, notice, the sample have different C content will product different amounts of CO2, and result in different pressure in the bottle), and the standard curve for the peak area—CO2 in the bottle (or the WSOC, but need do the conversion efficiency test first).

(7) How to explain the WSOC-IPMS method larger than the WSOC-TOC (fig 4)? And why some 30 $\mu$g-standard have the WSOC larger than 30 $\mu$g? Please reconsider (5) and (6).

(8) Line 206: the explain for the 13C-WSOC increase is kinetic isotope effect (KIE). This explanation is not reasonable because the leak process is fast and driven by strong motivation, which may not result in KIE. The mixture with CO2 in ambient air ( -9‰ to -7‰ seems more reasonable. In addition, I agree the leak-sample will have a low WSOC content, but not all the leak-sample have a large 13C value showed in fig 2, for example, there are one 30 $\mu$g standard sample have relatively lower WSOC content, but no significant difference between all the 30 $\mu$g samples. Please explain.

(9) The final point bothers me is there are relatively large differences ( 3‰ between the monocomponent standard samples with different WSOC content. Then for the complicated real aerosol samples, during each period, the 13C-WSOC changes ( 3‰ with WSOC content, how to evaluate the influence of WSOC content on the 13C-WSOC. If not evaluate, the conclusion for the source or transformation processes in each episode is lack of reliability

And there are some minor comments below point out areas that are not clear and make some suggestions.

Minor points:

Line 15-18 The difficulty for the method should be show

Line 23: give the real time resolution rather than using the "High time-resolved"

Line 33: not all the dust carbonates are water-insoluble carbon

Line 61: depleted, not enriched

Line 84: Reference

Line 89: Reference

Line 100-101: Reference, and show the representativeness for other standards.

Line 104: the exact $\delta$ for each standard, rather than the range

Line 106: give the C concentration (or the volume for the certain C mass) of the solution, and the exact magnitude, rather than the range.

Line: 145 the size of the filters

Line142: When was the filter in 2015 measured?

Line 168ïijŽthe carbon content unit?

Line 206: the sentence is not clear. Line 328-335 I agree that the fractionation during the secondary formation would result in the deplete of 13C in products (WSOC) compared to remaining substrate (the remaining VOCs), but how it can explain the explain the 13C enrichment in WSOC compared to TC ?

Line 716: Table 1 is not cited in the text

Line 702: add the size of each bottle, the volume and of solution, the flow rate of He. . . in Fig 1

Line 725: add the real $\delta$13C-KHP in Fig 2b, as a line

Line 728 and 738: Please unify the use of frame lines

Line 734: add the 1:1 line and the correlation line

2018.

---

## Author Comment (AC2) · 21 May 2019

The comment was uploaded in the form of a supplement:
https://www.atmos-chem-phys-discuss.net/acp-2018-1056/acp-2018-1056-AC2-supplement.pdf

---

## Author Comment (AC3) · 21 May 2019

The comment was uploaded in the form of a supplement:
https://www.atmos-chem-phys-discuss.net/acp-2018-1056/acp-2018-1056-AC3-supplement.pdf

---

## Author Response (AR1)

**Responses to the Anonymous Referee's comments**

**Referee #1 Evaluations:**

The manuscript reports an optimized method to analyze the $^{13}C$ in the WSOC with small amounts, then applied it in the high time resolution aerosol samples during the haze period, and try to explain the source and processes beyond the variations of $\delta^{13}C$-WSOC. Although the context of this study does not introduce a bran-new method to analyze the carbon isotope or the new atmospheric process beyond what has been known, the optimized method will be a valuable contribution to the knowledge of WSOC. Overall, the manuscript is organized and read well. The paper does have a number of major issues that need to be addressed before publication.

**Response:** We thank the reviewer for the nice summary of our paper and the positive assessment of this work. We have carefully revised the manuscript following the reviewer comments and suggestions. Our responses to all comments made by the reviewer are given below (in blue font). And the revised parts are also shown after the responses (in green font). Please refer to the revised manuscript, in which changes are highlighted in yellow.

**Major comments:**

(1) The manuscript set up an optimized method, and declare it is novel, but limited information for previous method (e.g. the theory, operation, use condition and deficiency...) present in the introduction section. Thus, the reader cannot get the challenge or difficulty for the establish of new method. Try to add the previous researches in the introduction, the challenge of the new method in the abstract, and the comparison in the discussion section.

**Response:** We agree with the reviewer; the information of the previous methods was added in the introduction part. The importance of the optimized method was briefly stated in the abstract. Please see line 18-20 and line 99-117 in the revised MS.

In the abstract, we added "However, the previous methods measuring the $\delta^{13}C$ values of WSOC in ambient aerosols require large amount of carbon contents as well as time-consuming and labor-intensive preprocessing."

In the introduction, we added "This is partially due to the limited techniques to analyze the $\delta^{13}C$ signatures of WSOC in ambient aerosols, as their concentrations are usually very small. In the recent years, some efforts have been made to measure the $\delta^{13}C$ values of WSOC. Bauer et al. (1991) uses potassium persulfate to convert organic carbon in natural waters into $CO_2$ for

$\delta^{13}$C measurements. This wet oxidation method requires more than 0.5mM C and 1h during the pretreatment (from sample injection to the isolation of purified $CO_2$). Fisseha et al. (2006) boiled the oxidizing solution for 45 min to remove the organic matter and the total time required for the pretreatment (for 15 samples) is 1.5h. Kirillova et al (2010) develops a combustion method that applies the aerosol extract without filtration for isotope measurement and involves complicated processes such as the freeze-drying of the aerosol extract under vacuum for 16 h. This combustion method is the most widely used for the $\delta^{13}$C measurements in WSOC aerosols (Kirillova et al., 2010, 2013, 2014; Miyazaki et al., 2012; Pavuluri et al, 2017). Although these methods are able to provide the $\delta^{13}$C values of WSOC in natural waters and/or ambient aerosols, the analytical methods require either large amount of WSOC (from 100 μg C to 0.5 mM C) or time-consuming preprocessing. And some of the methods oxidize the WSOC extract without filtration and/or decarbonation in the pretreatment., which would result in higher uncertainty of the $\delta^{13}$C results. The high detection limit of the previous methods is difficult to determine the $\delta^{13}$C-WSOC in aerosol samples with low carbon concentrations. In that case, an easily operated method detecting the $\delta^{13}$C-WSOC values in aerosol samples with low detection limit and high precision is urgently needed. ”

(2) Line 55-65: The isotope signatures of particulate matter (not the WSOC, or VOCs) emitted from different sources have limited the various of isotope signatures of $\delta^{13}$C-WSOC, due to the most part of the WSOC come from the secondary formation rather than the primary emission. In addition, the isotopic effects during the formation or aging process is absence of proof, need add some reference.

**Response:** Thanks for the reviewer's suggestion, the possible sources and the corresponding isotope compositions were introduced more clearly in the introduction. And the atmospheric processes and their isotopic effect were described in more detail with corresponding references (Atkinson R., 1986; Kirillova et al., 2013; Fisseha et al., 2009; Nina et al., 1979; Rudolph et al., 2002; Sakugawa and Kaplan, 1995, Fisseha et al., 2009; Pavuluri and Kawamura, 2012; Iannone et al., 2003; Rudolph et al., 2000; Anderson et al., 2004; Fisseha et al., 2009; Rudolph et al., 2003) in a separate paragraph. Please see line 73-94 in the revised MS.

We added "In addition, atmospheric processes like secondary formation and photochemical aging may change the constitution and properties of WSOC, as well as the stable carbon isotope of WSOC ($\delta^{13}$C-WSOC). According to the kinetic isotope effect (KIE), the reaction rate of molecules containing heavier isotopes is usually lower than the molecules containing lighter isotopes (Atkinson R., 1986; Kirillova et al., 2013; Fisseha et al., 2009). The change in reaction rate is primarily results from the greater energetic need for molecules containing heavier isotopes to reach the transition state (Nina et al., 1979). Consequently, the oxidants preferentially react with molecules with lighter isotopes (inverse kinetic isotope effect, KIE), which would result in an enrichment of $^{13}$C in the residual materials and a depletion in $^{13}$C of the particulate oxidation products (Rudolph et al., 2002). Therefore, organic compounds formed via secondary formation are generally depleted in $^{13}$C compared with their precursors (Sakugawa and Kaplan, 1995, Fisseha et al., 2009) and this isotope depletion is demonstrated in both field measurements and laboratory studies (Pavuluri and Kawamura, 2012). For example, the studies of KIE clearly indicate that the compounds formed via the oxidation are depleted in the $^{13}$C compared with their precursors during the reaction of VOCs with OH and ozone (dominant atmospheric oxidants) (Iannone et al., 2003; Rudolph et al., 2000; Anderson et al., 2004; Fisseha et al., 2009). Whereas an enrichment of $^{13}$C in the particulate organic aerosol may occur in the atmospheric aging processes, such as interactions with photochemical oxidants (e.g. hydroxyl radical and ozone) during the long range transport. For instance, studies have demonstrated that the substantial enrichment of $^{13}$C in the residual, aged aerosol (e.g. isoprene, a precursor of oxalic acid (Rudolph et al., 2003)) after a long range transport. In that case, the stable carbon isotope can be used to study the sources and the atmospheric processes that contribute to the carbonaceous aerosols.".

(3) The method needs to be explained much more clearly and in more detail, including the flow rate of He, the volume of the bottle (6 mL extraction for the 20 mm diameter disc, but how much for the TOC analysis, and how much use for the δ13C-WSOC analysis, if all use for δ13C-WSOC analysis, and another one for TOC, then the results in fig 4 need be reconsider due to the maldistribution in one filter), the volume of sample extracted solution, the ratio of sample extracted solution to the oxidizing agents....

**Response:** According to the reviewer's advice, the method part was described in more detail and more clearly in Section 2.4 in the revised MS. And some specific information was added in both text and figures (please see the response to the minor points).

In addition, 6 mL extraction for the 20 mm diameter disc was all for the analysis of the carbon content and $\delta^{13}$C values using Gas Bench II-IRMS. And another disc was dissolved for the analysis of TOC analyzer. The uniform distribution of the aerosol on the filter was a basic assumption in the aerosol filters analysis. In that case, the comparison of the results from the two equipment is reliable.

The volume of the sample extract was 4 mL, and 1mL oxidant solution made within 24 hours should be added in the vials. Please see Figure 1 and line 909-917 in the revised MS. The revised Section 2.4:

2.4 Sample pretreatment

The wet oxidation method is used to covert the WSOC to $CO_2$ (Sharp J. H., 1973), and the resulting $CO_2$ can be measured by IRMS. The overview of the optimized method for measuring WSOC and $\delta^{13}C_{-WSOC}$ in the aerosols is shown in Fig. 1. The process of the pretreatment consists of 6 steps: WSOC on a 20 mm diameter disc is extracted with 6 mL mili-Q water through water-bath ultrasonic for 30 minutes (step 1-2). The WSOC extract is filtered with a 0.22 μm syringe filter to remove the particles in step 3. 2.0 g potassium persulfate ($K_2S_2O_8$, Aladdin Industrial Corporation, Shanghai) and 100 μL phosphoric acid (85 % $H_3PO_4$, AR, ANPEL Laboratory Technologies Inc., Shanghai) are dissolved in 50 mL Milli-Q water to make the oxidizing solution. The oxidizing solution made within 24 h is added into the filtered WSOC extract as shown in step 4 of Fig. 1. The phosphoric acid is added to remove the inorganic carbon resolved in the solution, and the persulfate is added for the preparation to convert the organic compounds to $CO_2$. The vails are sealed tightly with the caps as soon as the oxidizing solution is added into the WSOC extract.

To remove the ambient $CO_2$ dissolved in the mixture (mixed solution of the oxidizing solution and the WSOC extract) and the atmospheric $CO_2$ in the headspace of the sealed sample vials, high-purity helium (Grade 5.0, 99.999 % purity) is flushed into the vials for 5 min in step 5. The aim of this step is to exclude the possible contamination from the atmospheric $CO_2$, and it has to be finished within 12 hours after the mixture of the WSOC extract and the oxidizing solution to avoid the loss of $CO_2$ produced under room temperature. High-purity helium (15-18 mL min$^{-1}$) is flushed under the water surface and a stainless steel tube is set for the output gas stream. The open end of this tube is submerged in Milli-Q water to prevent any backflow of atmospheric $CO_2$ (Fig. 1., step 5). After flushing, the vials are heated at 100 ˚C for 60 min in the sand bath pot (quartz sand, Y-2, Guoyu, China) to start the oxidation of WSOC in step 6. The heated vials are stored overnight at room temperature for condensing the moisture before the analysis on IRMS to prevent the damage to the measuring equipment.

[Figure]

**Figure 1.** Schematic of the optimized method for the measurement of WSOC mass concentrations and the $\delta^{13}C_{-WSOC}$ values. (A filter disc is dissolved with 6mL Mili-Q water in a 20 mL pre-combusted glass bottle in the first step. After 30 minutes autrasonic bath, the WSOC extract is filtered with 0.22 μm syringe filter and transferred to another 20 mL pre-combusted glass bottle in step 3. 4 mL filtrate is transferred to a 12 mL pre-combusted glass vial which contains 1 mL oxidant solution (2.0g $K_2S_2O_8$ and 100 μL 85% $H_3PO_4$ dissolved in 50 mL Mili-Q water) in the vial in step 4. Next, the mixed solution of WSOC extract and the oxidant solution is flushed with Helium at a flow rate of 15-18 mL min$^{-1}$ as shown in step 5. At last, the vials are heated for 60 minutes under 100 ℃ in the sand bath pot (step 6).)

(4) adjust the 2.4 and 2.5 section after 2.1 section.

**Response:** Following the reviewer's suggestion, the sections were adjusted as the advised order.

(5) The conversion efficiency (removal efficiency) showed be test, which is important to quality assurance of the δ13C-WSOC results. That is because there is fractionation during the conversion processes if the conversion efficiency is low, especially during a long reaction time. The operations of removal efficiency should cover two aspects: the residual WSOC in the reaction solution and the amount of $CO_2$ in the bottle after heating, and compare them with the added WSOC.

**Response:** In order to illustrate the method that we qualify the carbon content in unknown samples, we added a new section. Please see section 3.5.1 (line 298-317) in the revised MS.

We tested the peak areas of $CO_2$ gas and KHP solutions (results shown in Fig. S2.). Mixture gas of $CO_2$ and He (0.3% mol/mol) was injected into the vials (containing carbon content from 1ug to 24 ug) for the peak area measurement. Then a standard curve of peak areas Vs carbon contents in $CO_2$ gas (Carbon content (μg) = Peak area (Vs) × (2.50 ± 0.08) − (0.62 ± 0.39), $R^2$=0.98) was set up as Fig. S2. Similarly, another standard curve of peak areas Vs carbon contents in KHP solution can be established (from the results of KHP) as well (Carbon content (μg) = Peak area (Vs) × (2.34± 0.01) − (0.86±0.14), $R^2$=1.00), Fig. S2).

Both of the carbon contents in $CO_2$ gas and the KHP solution showed strongly linear correlation with peak areas. The slopes of the two linear equations were close to each other. The similarity of the two standard curves demonstrated that the WSOC extract can be oxidized to $CO_2$ gas containing similar amount of carbon. For example, KHP solution containing 10 ug C was expected to show peak area about 4.64 Vs, reflecting 10.8 ug C of $CO_2$ gas in the inflow of IRMS. Then the conversion efficiency was roughly calculated (carbon content in $CO_2$ gas / input carbon content in the WSOC extract) to be 108%. This represented that almost all the WSOC extract was converted to $CO_2$ by the oxidation. Then we assumed the WSOC extract was completely converted to $CO_2$ and would show no significant isotope fractionation in our method.

The revised section 3.5.1:

**3.5.1 Quantification of the carbon content**

The sample peak area is proportional to the carbon content in the vial and then is used to quantify the amount of $CO_2$ in the inflow of IRMS. The average value of the peak areas for the last eight sample peaks is taken as the peak area of a certain sample. The first two sample peaks are excluded to avoid the effect of the residual $CO_2$ of the former vial. We established a carbon content standard curve (linear equation) by measuring the peak areas of $CO_2$ gas samples containing 1-24 µg C (Fig. S2.). It has to be noted that the gas samples containing larger carbon contents are not tested for the difficulty of injecting too much volume of $CO_2$/He gas. Then the amount of $CO_2$ oxidized from the unknown samples can be quantified with this linear equation (i.e., Carbon content (µg) = Peak area (Vs) × (2.50 ± 0.08) − (0.62 ± 0.39), $R^2$=0.98). The standard curve (linear equation) of the peak areas against the carbon contents in the WSOC solution (KHP solution containing 1-100 µg C) is also established (Fig. S2.). And a linear equation similar with the peak areas against $CO_2$ gas is obtained (i.e., Carbon content (µg) = Peak area (Vs) × (2.34±0.01) − (0.86±0.14), $R^2$=1.00).

Then the conversion efficiency of the WSOC extract containing 1-100 µg C can be roughly calculated as 104 ±3 %. The high conversion efficiency demonstrates the completely conversion and the negligible isotope fractionation during the oxidation. In that case, the carbon content in the WSOC extract of unknown samples can be calculated based on the standard curve of peak areas against the carbon content in the WSOC extract. And the standard curve quantifying the carbon content has to be established with every batch of unknown samples to assure the completely conversion.

[Figure]

Figure S2. Standard curve to quantify the unknown samples.

(The standard curve is established by the $CO_2$ gas / the KHP solution and the input carbon content of the certain vials. The blue dotted line is the linear fit of the results of $CO_2$ gas, and the black dotted line is the linear fit of the results of KHP solution.)

(6) The IRMS method for WSOC concentration showed be present detail in the method section, rather than a single sentence like "Line 236 the peak area obtained from the measurement could represent the carbon content". If it means using the peak area represent the CO2 amounts in the inflow of IRMS? If yes, then show which peak use (10 peak total?), the gas volume in bottle and into IRMS (in the atm pressure, notice, the sample have different C content will product different amounts of CO2, and result in different pressure in the bottle), and the standard curve for the peak areaâ ˘A ˘TCO2 in the bottle (or the WSOC, but need do the conversion efficiency test first).

**Response:** Following the reviewer's suggestion, the IRMS method for WSOC concentration were described in more detail in a new section 3.5.1. Please see the response to the last comment (5), or line 298-317 in the revised MS.

The peak areas obtained from the Gas Bench II-IRMS were proportional to the carbon contents in the vials. Thus the peak areas were taken to represent the carbon content. We used the average values of the last 8 peaks to avoid the memory effect in the first two peaks. The gas volume in the bottles were the same (7 mL of headspace in the vials), and 100 μL $CO_2$ mixed with He was injected into the IRMS every 70 s. The standard curve of the peak areas and the carbon contents (amount of $CO_2$) in the vials was Carbon content (μg) = Peak area (Vs) $\times (2.34 \pm 0.01) - (0.86 \pm 0.14)$, $R^2 = 1.00$). It was added in the supporting information (S2). Please see the response to (5).

(7) How to explain the WSOC-IPMS method larger than the WSOC-TOC (fig 4)? And why some 30 μg-standard have the WSOC larger than 30 μg? Please reconsider (5) and (6)

**Response:** There is systematic bias between the two equipment (TOC analyzer and Gas Bench II), and the recovery of the ambient samples is $99 \pm 10\%$. In addition, data shown in Fig. 4. were the results of ambient WSOC aerosols instead of the standards. The data were obtained from Gas Bench II-IRMS (x axis) and TOC analyzer (y axis). Also, the carbon contents were calculated with the peak areas in the wet oxidation method using the Gas Bench II – IRMS. According to the recovery ($97 \pm 6\%$) of this method, we regarded the WSOC conversion was complete, and the carbon content measured using this optimized method was reliable.

(8) Line 206: the explain for the 13C-WSOC increase is kinetic isotope effect (KIE). This explanation is not reasonable because the leak process is fast and driven by strong motivation, which may not result in KIE. The mixture with CO2 in ambient air ( -9‰ to -7‰ seems more reasonable. In addition, I agree the leak-sample will have a low WSOC content, but not all the leak-sample have a large 13C value showed in fig 2, for example, there are one 30 μg standard sample have relatively lower WSOC content, but no significant difference between all the 30 μg samples. Please explain.

**Response:** We agree that the gas leak of the vials with deformed caps are fast and driven by strong motivation. But gases in the vials could also leak out through the tiny holes purged by the stainless tube in the helium flushing step. And this kind of leak is slow and not easy to be observed.

In our opinion, atmospheric $CO_2$ was not a possible reason of the phenomenon we observed (lower carbon content corresponding with higher isotope composition). Atmospheric $CO_2$ contamination would result in higher carbon content and higher isotopic ratios according to the isotopic ratio of $CO_2$ in ambient air (-9‰ to -7‰). But this was not observed in our results. In addition, few procedures were processed to exclude the effect of the atmospheric $CO_2$ in the pretreatment (i.e., Step 4). $H_3PO_4$ was added in the oxidant solution to remove the atmospheric $CO_2$ dissolved in the WSOC extract as well (Step 5). Then the high purity helium was flushed into the vials to remove the $CO_2$ in ambient air in the headspace of the vials. In addition, the peak areas of the Mili-Q water (which has been processed with all the procedures in the pretreatment) tested in Section 3.1 were not detected, demonstrating negligible effect of the atmospheric $CO_2$ in the measurement of $CO_2$.

But we actually should apologize that the Fig. 2. was drawn in the wrong range of y axis which resulted in the loosing information of the dramatically high delta values corresponding with the low carbon contents. The revised graph was given below and the it could be seen in line 918 in the revised MS.

[Figure]

Figure 2. Carbon contents (a) and isotopic ratios (b) of KHP after different heating time.

(9) The final point bothers me is there are relatively large differences (3‰ between the monocomponent standard samples with different WSOC content. Then for the complicated real aerosol samples, during each period, the 13C-WSOC changes (3‰ with WSOC content, how to evaluate the influence of WSOC content on the 13C-WSOC. If not evaluate, the conclusion for the source or transformation processes in each episode is lack of reliability.

**Response:** From Fig. 3., we can see the corrected delta values of KHP (SD=0.14, n=15) and BA (SD=0.15, n=10) were stable (regardless of the carbon content in the standard) and close to the correct values ($\delta^{13}C_{-KHP}$ = -30.40‰, $\delta^{13}C_{-BA}$ = -27.17‰) tested by EA-IRMS. But the isotopic ratios of $CH_6$ and $C_2$ were more variable at small amount of carbon contents (i.e., <5μg C). We agree that the analysis of isotope compositions is not reliable if the standards show difference larger than 1‰ (SD >0.5‰) in delta values. The standard deviations of isotope results of each standard are better than 0.17 ‰ when the carbon contents are larger than 5 μg C. In that case, the detection limit of this method was 5 μg C and the results (both carbon contents and the isotopic ratios) of WSOC lower than 5 μg C were not reliable (for now). And it has to be noticed that the standard deviation of the isotope results of $CH_6$ was calculated without one exception ($CH_6$ at 6 μg C), the corrected isotopic ratio was -13.6 ‰, still about 1.5 ‰ difference with other repetitions. It was probably a wrong measurement according to the stable isotopic ratios, then it was taken as an outlier of the results.

[Figure]

Figure 3. Isotope results before and after the two-step correction of the four standards. (a. KHP, b. BA, c. $CH_6$, d. $C_2$. Red circle with a spot represents the two-step corrected isotopic ratios; ■, ◆, △, × represent the raw data from Gas Bench II; the dotted line represents the blank corrected $\delta^{13}C$ values tested by EA)

**Minor points:**

Line 15-18 The difficulty for the method should be show

**Response:** We agree with the reviewer, and the importance and the difficulty of measuring small amount of WSOC in aerosol samples were briefly introduced in the abstract. Please see line 18-20 in the revised MS.

We added "However, the previous methods measuring the $\delta^{13}C$ values of WSOC in ambient aerosols require large amount of carbon contents as well as time-consuming and labor-intensive preprocessing."

Line 23: give the real time resolution rather than using the "High time-resolved"

**Response:** According to the reviewer's advice, we gave the real time solution (i.e., the aerosol samples collected every 3 hours) here. Please see line 25 in the revised MS.

The revised sentence:

However, the previous methods measuring the $\delta^{13}C$ values of WSOC in ambient aerosols require large amount of carbon contents as well as time-consuming and labor-intensive preprocessing.

Line 33: not all the dust carbonates are water-insoluble carbon

**Response:** We agree with the reviewer, even though a large fraction of dust carbonate is water-insoluble. So we changed the incorrect statement ("water-insoluble carbon") to "non-WSOC fraction". Please see line 35 in the revised MS.

The revised sentence:

This suggests that non-WSOC fraction in total carbon may contain $^{13}C$-enriched components such as dust carbonate which is supported by the enhanced $Ca^{2+}$ concentrations and air mass trajectories analysis.

Line 61: depleted, not enriched

**Response:** "Depleted" and "enriched" are relative words, for instance, marine aerosols are enriched in $^{13}$C compared with C3 plant biomass burning, but depleted in $^{13}$C compared with C4 plant biomass burning. So we replaced the saying of "depleted" or "enriched" with actual isotope composition to avoid the misunderstanding. Please see line 67 in the revised MS.

The revised sentence:

Marine organic aerosol sources have a carbon isotope signature of -22 ‰ to -18 ‰, (Miyazaki et al., 2011) and play an important role in the aerosols at coastal sites.

Line 84: Reference
Line 89: Reference

**Response:** References were added, please see line 98 and 116 in the revised MS.

Added references:

(Fisseha et al., 2006; Kirillova et al., 2010; Suto et al., 2018; Lang et al, 2012; Zhou et al., 2015).

Line 100-101: Reference, and show the representativeness for other standards.

**Response:** According to the reviewer's suggestion, we added the references and the representativeness for other standards. Please see line 128-131 in the revised MS.

The revised sentence:

KHP and BA are widely used as the standards of WSOC measurements (Kirillova et al., 2010) and then are used here as the WSOC test substances. Also, their isotope signatures are close to the $\delta^{13}$C values of aerosol samples (Miyazaki et al., 2012; Fisseha et al., 2009; Suto et al., 2018).

Line 104: the exact δ for each standard, rather than the range

**Response:** Following the reviewer's suggestion, the exact delta values of each standard were given in both text (line 133) and Figure 3 in the revised MS.

We added "The carbon isotope composition of these four standards are: -12.20 ‰ ($CH_6$), -13.84 ‰ ($C_2$), -27.17 ‰ (BA) and -30.04 ‰ (KHP), respectively."

Line 106: give the C concentration (or the volume for the certain C mass) of the solution, and the exact magnitude, rather than the range.

**Response:** Following the reviewer's suggestion, the exact concentration of each standard was given in line 136-139 in the revised MS.

We added "Standards are resolved in Milli-Q water (18.2 MΩ quality) to make standard solutions of 0.25μg mL$^{-1}$, 0.75 μg mL$^{-1}$, 1.5μg mL$^{-1}$, 3μg mL$^{-1}$, 6μg mL$^{-1}$, 12μg mL$^{-1}$ and 24μg mL$^{-1}$, which means containing carbon content of 1ug, 3ug, 6ug, 12ug, 24ug, 48ug and 96ug in 4mL standard solution to test the procedures during the pretreatment."

Line: 145 the size of the filters

**Response:** The size of the filter is 180×230mm, please see line 145 in the revised MS.

We added "PM$_{2.5}$ samples are collected on pre-combusted quartz-fiber filters (180×230mm) every 3 hours with a high-volume aerosol sampler (KC100, Qingdao, China) at a flow rate of 1 m$^3$ min$^{-1}$." in the revised MS.

Line142: When was the filter in 2015 measured?

**Response:** The WSOC on the filter collected in 2015 was measured in Nov 2016.

Line 168ïijŽthe carbon content unit?

**Response:** The unit of the carbon content was microgram (μg).

Line 206: the sentence is not clear.

**Response:** We have rewritten this sentence. Please see line 256-260 in the revised MS.
The revised sentence:

According to the kinetic isotope effect (KIE), isotope fractionation occurs during the gas leaking. The light carbon isotopes ($^{12}$C) are easier to escape from the vials than the heavy ones ($^{13}$C), thus the remaining CO$_2$ would be more enriched with heavy isotopes ($^{13}$C). In that case, lower carbon contents and higher $\delta^{13}$C values are expected to be observed in the results of leaking vials.

Line 328-335 I agree that the fractionation during the secondary formation would result in the deplete of 13C in products (WSOC) compared to remaining substrate (the remaining VOCs), but how it can explain the explain the 13C enrichment in WSOC compared to TC?

**Response:** This paragraph here was explaining the secondary formed WSOC tend to be depleted in $^{13}$C. Thus the $^{13}$C enrichment in WSOC compared to TC **was not** mainly affected by the secondary formation of WSOC.

Line 716: Table 1 is not cited in the text

Response: We cited Table 1 in line 206 in the revised MS.

We added "To achieve this goal, the procedural blanks are analyzed to test the contamination that the reagents would introduce to the results (shown in Table 1). "

Line 702: add the size of each bottle, the volume and of solution, the flow rate of He...in Fig 1

Response: Following the reviewer's advice, we added the information in the description of Fig. 1. Please see line 909-917 in the revised MS.

The revised description of Fig.1.:

Figure 1. Schematic of the optimized method for the measurement of WSOC mass concentrations and the $\delta^{13}C_{-WSOC}$ values. (A filter disc is dissolved with 6mL Mili-Q water in a 20 mL pre-combusted glass bottle in the first step. After 30 minutes autrasonic bath, the WSOC extract is filtered with 0.22 μm syringe filter and transferred to another 20 mL pre-combusted glass bottle in step 3. 4 mL filtrate is transferred to a 12 mL pre-combusted glass vial which contains 1 mL oxidant solution (2.0g $K_2S_2O_8$ and 100 μL 85% $H_3PO_4$ dissolved in 50 mL Mili-Q water) in the vial in step 4. Next, the mixed solution of WSOC extract and the oxidant solution is flushed with Helium at a flow rate of 15-18 mL min$^{-1}$ as shown in step 5. At last, the vials are heated for 60 minutes under 100 ℃ in the sand bath pot (step 6).)

Line 725: add the real δ13C-KHP in Fig 2b, as a line

Response: Following the reviewer's advice, we added the correct value of $\delta^{13}C$-KHP in Fig. 2b. as a line. Please see figure 2b. in the response to (8) or line 919 in the revised MS.

Line 728 and 738: Please unify the use of frame lines

Response: Following the reviewer's advice, we added the frame lines in Fig. 3. and 5. Please see Fig. 3. in the response to (9) or line 922 in the revised MS.

The revised Fig. 5.: line 931 in the revised MS.

[Figure]

**Figure 5.** Time series of PM$_{2.5}$, WSOC, precipitation, $\delta^{13}$C values, nss-K$^+$, Ca$^{2+}$, TC, wind speed and wind direction at the sampling site during the studied period. (The time period framed with the rectangles is defined as the Episode 1 (green), the Episode 2 (red) and the Episode 3 (orange). The dotted line in 5e is the average value of nss-K$^+$ during the studied period. The high concentration and intense increase of nss-K$^+$ in the Episode 2 indicate a significant biomass burning (BB) event, and is marked with "BB event" in 5e. The similar trends of Ca$^{2+}$ and TC suggest a dust event in the Episode 3.)

Line 734: add the 1:1 line and the correlation line

**Response:** Following the reviewer's advice, we added the 1:1 line and the correlation line as well as the linear equation in Figure 4. Please see line 927 in the revised MS.

The revised Fig.4.:

[Figure]

**Figure 4.** Correlation of WSOC mass concentrations measured with Gas Bench II - IRMS and TOC analyzer.

**Referee #2 Evaluations:**

The manuscript shows the setup and thorough evaluation of a method to analyze 13C in small amounts of water-soluble organic carbon (WSOC). This method is then applied in an interesting case study and demonstrates the high time resolution that can be achieved by analyzing such small sample amounts. Measuring 13C accurately and precisely on aerosol samples as small as 5 μg is challenging and it is nice to see that the authors succeed in this.

**Response:** Thanks for the careful reading and valuable suggestions to improve the scientific content of the manuscript. Following the comment of the reviewer, we have carefully made corrections in the revised MS. Our responses to all comments made by the reviewer are given below (in blue font). And the revised parts are also shown after the responses (in green font). Please refer to the revised MS, in which changes are highlighted in yellow.

**General comments**

(1) part of the system testing and of the analysis procedure needs to be explained much more clearly and in more detail, before it is really understandable (see specific comments), ie. Probably 1-2 additional figures are required for the supporting material

**Response:** Following the reviewer's advice, we explained the analysis procedures more clearly and in more detail, and we added few figures in the revised supporting information (i.e., S2-S4). Please see the response to Referee #1.

(2) All the reproducibility tests were done with standard only, but the reproducibility of actual filter samples is usually lower and this should be tested as well. Please select a couple filters (with high and low loading) and analyze them repeatedly to get a better idea of reproducibility for WSOC extracted from filter samples

**Response:** The reproducibility of the actual aerosol samples was tested by analyzing them repeatedly. The reproducibility of the WSOC concentration and the isotope results were 0.25 ± 0.04 μg C and 0.14 ± 0.07 ‰ respectively, and were given in section 3.6 in the revised MS. The reproducibility was the average value of the standard deviation of the results (WSOC concentration and the isotopic ratios).

We added the reproducibility as: "The ambient aerosol filters are tested repeatedly to evaluate the reproducibility of the ambient samples as well. The standard deviation of the

WSOC concentrations and the isotope results of the repeated ambient samples are 0.25 ±0.04 μg C (n≥3) and 0.14 ± 0.07 ‰ (n≥3), respectively."

(3) The writing clarity and the grammar need to be improved, see specific comments for examples, but try to proofread the whole manuscript

Response: Thank you very much for the suggestion to refine the language, we proofread the whole manuscript and revise the language that may bring confusion to the readers.

**Specific comments:**

Line 47/48: In don't understand what this sentence means and it seems out of context. What is meant by "surface layer" and does the transport refer to vertical transport (e.g. deposition) or some other transport, such as long-range transport? Unless you explain in much more detail, omit this sentence

Response: Following the reviewer's advice, this sentence was deleted in the revised MS.

Line 61: Please provide reference with respect to fractionation during biomass burning

Response: References were added in line 64-67, please see in the revised MS.

We added "Laboratory studies demonstrate that there is no significant isotope fractionation (± 0.5 ‰, Turekian et al., 1998; Currie et al., 1999) compared to the plant material during the biomass burning of C3 plants (Das et al., 2010). While the C4 plants combustion results in $^{13}$C depletion (<0.5 to 7.2%) in the produced aerosols (Turekian et al., 1998; Das et al., 2010)."

Line 69-78: Please explain this in a bit more detail and avoid unclear formulations, e.g. "lighter isotopes (12C) have the priority to be oxidized", eg mean that "molecules containing lighter isotopes usually react faster than molecules containing heavier isotopes" (and this is still an oversimplification, but more acceptable than what is written here; "isotope depleted matters": The matter is not depleted in isotopes, it is depleted in the heavy isotope (13C), "secondary formed WSOC" = WSOC formed by secondary processes, "positive isotopic fractionation" = I think you mean enrichment here?

Response: According to the reviewer's suggestion, we rewrote this part, and explained the fractionation effect in more detailed language to exclude the use of unclear formulations. Please see line 73-94 in the revised MS.

The revised paragraph:

In addition, atmospheric processes like secondary formation and photochemical aging may change the constitution and properties of WSOC, as well as the stable carbon isotope of WSOC ($\delta^{13}C_{-WSOC}$). According to the kinetic isotope effect (KIE), the reaction rate of molecules containing heavier isotopes is usually lower than the molecules containing lighter isotopes (Atkinson R., 1986; Kirillova et al., 2013; Fisseha et al., 2009). The change in reaction rate is primarily results from the greater energetic need for molecules containing heavier isotopes to reach the transition state (Nina et al., 1979). Consequently, the oxidants preferentially react with molecules with lighter isotopes (inverse kinetic isotope effect, KIE), which would result in an enrichment of $^{13}C$ in the residual materials and a depletion in $^{13}C$ of the particulate oxidation products (Rudolph et al., 2002). Therefore, organic compounds formed via secondary formation are generally depleted in $^{13}C$ compared with their precursors (Sakugawa and Kaplan, 1995, Fisseha et al., 2009) and this isotope depletion has proven by both field measurements and laboratory studies (Pavuluri and Kawamura, 2012). For example, the studies of KIE clearly indicate that the compounds formed via the oxidation are depleted in the $^{13}C$ compared with their precursors during the reaction of VOCs with OH and ozone (dominant atmospheric oxidants) (Iannone et al., 2003; Rudolph et al., 2000; Anderson et al., 2004; Fisseha et al., 2009). Whereas an enrichment of $^{13}C$ in the particulate organic aerosol may occur in the atmospheric aging processes, such as interactions with photochemical oxidants (e.g. hydroxyl radical and ozone) during the long range transport. For instance, studies have demonstrated that the substantial enrichment of $^{13}C$ in the residual, aged aerosols (e.g. isoprene, a precursor of oxalic acid (Rudolph et al., 2003) after a long range transport. In that case, the stable carbon isotope can be used to study the sources and the atmospheric processes that contribute to the carbonaceous aerosols.

Line 74: what does "these" refer to.

Please reformulate this entire paragraph and use as precise and correct language as

Possible

**Response:** Following the reviewer's suggestion, we reformulated the sentences and illustrated the fractionation effect of the secondary formation process and the photochemical aging process of the WSOC in ambient aerosols. "These" here actually referred to the isotopic fractionation effect resulted by the various atmospheric processes like secondary formation and aging process. Please see line 99-117 in the revised MS.

The revised sentences:

This is partially due to the limited techniques to analyze the $\delta^{13}C$ signatures of WSOC in ambient aerosols, as their concentrations are usually very small. In the recent years, some efforts have been made to measure the $\delta^{13}C$ values of WSOC. Bauer et al. (1991) uses potassium persulfate to convert organic carbon in natural waters into $CO_2$ for $\delta^{13}C$ measurements. This wet oxidation method requires more than 0.5mM C and 1h during the pretreatment (from sample injection to the isolation of purified $CO_2$). Fisseha et al. (2006) boiled the oxidizing solution for 45 min to remove the organic matter and the total time required for the pretreatment (for 15 samples) is 1.5h. Kirillova et al (2010) develops a combustion method that applies the aerosol extract without filtration for isotope measurement and involves complicated processes such as the freeze-drying of the aerosol extract under vacuum for 16 h. This combustion method is the most widely used for the $\delta^{13}C$ measurements in WSOC aerosols (Kirillova et al., 2010, 2013, 2014; Miyazaki et al., 2012; Pavuluri et al, 2017). Although these methods are able to provide the $\delta^{13}C$ values of WSOC in natural waters and/or ambient aerosols, the analytical  methods require either large amount of WSOC (from 100 μg C to 0.5 mM C) or time-consuming preprocessing. And some of the methods oxidize the WSOC extract without filtration and/or decarbonation in the pretreatment., which would result in higher uncertainty of the $\delta^{13}C$ results. The high detection limit of the previous methods is difficult to determine the $\delta^{13}C_{-WSOC}$ in aerosol samples with low carbon concentrations. In that case, an easily operated method detecting the $\delta^{13}C_{-WSOC}$ values in aerosol samples with low detection limit and high precision is urgently needed.

Line 89: This sentence seems out of context here, please explain this in more detail in a separate paragraph

**Response:** This sentence was deleted because the meaning of this sentence does not fit the main idea in this paragraph.

Paragraph 2.2: The description is not clear. The figure itself quite clear and informative, but the text is not. The figure contains several steps, it would help to make clear in the text, which step (e.g., 1, 2,3 . . .) you are talking about. I think for clarity, you should refer to the extract from the filters as "WSOC extract" instead of extraction. After you add the oxidizing solution you name the resulting mixture sometimes as "mixture", sometimes as "solution", this is confusing – choose one. Make clear that the removal of CO2 from the mixture refers to dissolved ambient CO2 (i.e. a cleaning step before oxidation). People, who don't know the subject well might assume that just adding the oxidizing solution would lead to oxidation.

Finally make clear (in the text) that heating in the sand bath results in the oxidation of WSOC to CO2.

**Response:** Following the reviewer's suggestion, we have rewritten this paragraph. We introduced the steps in more detail and mentioned the exact steps (as step 1, 2 …) in the text. We replaced the ambiguous terms as suggested "WSOC extract", and we used "mixture" when we were talking about the mixed solution of WSOC extract and oxidizing solution. Also, we mentioned that the He flushing step was to remove the $CO_2$ dissolved in the solution and the atmospheric $CO_2$ in the headspace. In addition, we illustrated that it was the heating in the sand bath resulted in the oxidation of WSOC to $CO_2$. Please see line 157-181 in the revised MS.

The revised Section:

2.4 Sample pretreatment

The wet oxidation method is used to covert the WSOC to $CO_2$ (Sharp J. H., 1973), and the resulting $CO_2$ can be measured by IRMS. The overview of the optimized method for measuring WSOC and $\delta^{13}C_{-WSOC}$ in the aerosols is shown in Fig. 1. The process of the pretreatment consists of 6 steps: WSOC on a 20 mm diameter disc is extracted with 6 mL mili-Q water through water-bath ultrasonic for 30 minutes (step 1-2). The WSOC extract is filtered with a 0.22 μm syringe filter to remove the particles in step 3. 2.0 g potassium persulfate ($K_2S_2O_8$, Aladdin Industrial Corporation, Shanghai) and 100 μL phosphoric acid (85 % $H_3PO_4$, AR, ANPEL Laboratory Technologies Inc., Shanghai) are dissolved in 50 mL Milli-Q water to make the oxidizing solution. The oxidizing solution made within 24 h is added into the filtered WSOC extract as shown in step 4 of Fig. 1. The phosphoric acid is added to remove the inorganic carbon resolved in the solution, and the persulfate is added for the preparation to convert the organic compounds to $CO_2$. The vails are sealed tightly with the caps as soon as the oxidizing solution is added into the WSOC extract.

To remove the ambient $CO_2$ dissolved in the mixture (mixed solution of the oxidizing solution and the WSOC extract) and the atmospheric $CO_2$ in the headspace of the sealed sample vials, high-purity helium (Grade 5.0, 99.999 % purity) is flushed into the vials for 5 min in step 5. The aim of this step is to exclude the possible contamination from the atmospheric $CO_2$, and it has to be finished within 12 hours after the mixture of the WSOC extract and the oxidizing solution to avoid the loss of $CO_2$ produced under room temperature. High-purity helium (15-18 mL min$^{-1}$) is flushed under the water surface and a stainless steel tube is set for the output gas stream. The open end of this tube is submerged in Milli-Q water to prevent any backflow of atmospheric $CO_2$ (Fig. 1., step 5). After flushing, the vials are heated at 100 ˚C for 60 min in the sand bath pot (quartz sand, Y-2, Guoyu, China) to start the oxidation of WSOC in step

6. The heated vials are stored overnight at room temperature for condensing the moisture before the analysis on IRMS to prevent the damage to the measuring equipment.

Line 126: What gases are typically in the headspace after oxidation?

**Response:** After oxidation, the gases in the headspace would be $CO_2$ mixed with helium. This was added in line 183 in the revised MS.

The revised sentence:

$CO_2$ gas produced in the headspace of the prepared sample is extracted and purified by Gas Bench II (Gas Bench II, Thermo Fisher Scientific, Bremen, Germany), and introduced into an isotope ratio mass spectrometer (IRMS) (Mat 253, Thermo Fisher Scientific, Bremen, Germany) for $\delta^{13}C_{-CO2}$ analysis.

Line 159: This sentence might fit better in the method section

**Response:** Following the reviewer's suggestion, we moved this sentence to the method section, please see the Section 2.4 in the revised MS or the response to the former comments.

Line 164: Here you speak about "oxidizing agents" in plural, but in the text only one (H3PO4), is discussed. What about the other components of the solution?

**Response:**

We actually discussed the contamination of each component in the oxidant solution (Mili-Q water, $H_3PO_4$ and $K_2S_2O_8$). And to avoid the misunderstanding, we changed the title of this section to "The carbon content in the procedural blank", please see section 3.1 in the revised MS.

We found the carbon content in Mili-Q water was not detected. Then we discussed the use of different kinds of $H_3PO_4$, and found no significant difference of carbon contents between 85% $H_3PO_4$, with carbon content of 0.03-0.04 µg C. And after adding the oxidant in to the procedural blank, the carbon contents increased to about 0.5 µg C. In that case, the major contamination in the procedural blank came from the oxidant.

The revised Section 3.1:

3.1 The carbon content in the procedural blank

In order to quantify the low concentration of WSOC in aerosols, it is critical to reduce the carbon content in the procedural blank for minimizing the detection limit of the method. To achieve this goal, the procedural blanks are analyzed to test the contamination that the reagents would introduce to the results (shown in Table 1). The average carbon content in the procedural blank is about 0.5 μg C (corresponding with a peak area of 0.23 Vs) with a $\delta^{13}C$ value of -27.04 ± 1.28 ‰ (n=15). The carbon contents and the isotope compositions of Mili-Q water and the agents dissolved in the oxidizing solution are also determined to identify the source of contamination in the procedural blank. The peak area of Mili-Q water is not detected (Table. 1.) after going through all the processes in the pretreatment without adding any other materials, suggesting no contamination is introduced from the Mili-Q water. After that, the contamination from 85% $H_3PO_4$ with different purity (acid-1: analytical reagent, AR; acid-2: High Performance Liquid Chromatography, HPLC) are compared. The carbon contents in the 85% $H_3PO_4$ dissolved in Mili-Q water are 0.03-0.04 μg C and show no significant discrepancy between different purity.

Interestingly, the carbon content increase to 0.5-0.6 μg C after the persulfate is added, implicating that the $CO_2$ in the procedural blank is mainly produced from the oxidation of organic substance in the persulfate. The carbon content in HPLC grade of 85% $H_3PO_4$ mixed with the persulfate (0.58 - 0.63 μg C) is closed to that of AR grade (0.46 - 0.63 μg C, see table 1.). Thus, AR grade with purity of 85 % $H_3PO_4$ is utilized to prepare the oxidizing solution in this method. The average carbon content of the procedural blank is estimated to be 0.5 ±0.06 μg C, and the detection limit is expected to be 10 times the procedural blank (i.e. 5 μg C). The carbon content in the procedural blank of this method is much lower than that of the methods analyzing isotopes of WSOC in aquatic environment or soil (De Groot, 2004; Polissar et al., 2009; Werner et al., 1999). The smaller carbon content of the procedural blank suggests the possibility to correctly measure the WSOC and $\delta^{13}C_{-WSOC}$ of samples containing low carbon content.

Line 167: "blank effect of H3PO4" unclear formulation – I think you mean that you determine the carbon content of H3PO4, which will introduce contamination (blank) in your analysis.

**Response:** Following the reviewer's suggestion, "blank effect of $H_3PO_4$" was revised as "the contamination that the reagents would introduce to the results" in line 205 of the revised MS.

Line 170-172: To make this more clear state the procedural blank (average +- std) before and after addition of H3PO4

**Response:** The contaminations of the procedural blanks were not detected before adding the $H_3PO_4$, and after the addition of $H_3PO_4$, the procedural blank varied between 0.03-0.04 µg C. This information was included in table 1. Please see Table 1 (line 905) in the revised MS.

**Table 1.** Various blank preparation with results.

| Identifier | Oxidant [a] | Acid [b] | C content (µgC) | $\delta^{13}C$(‰) |
|---|---|---|---|---|
| Mili-Q water | - | - | ND* | - |
| Mili-Q water | - | - | ND* | - |
| Mili-Q water +Acid-1 | - | 100 uL 85 % $H_3PO_4$, AR | 0.04 | -1.6 |
| Mili-Q water +Acid-1 | - | 100 uL 85 % $H_3PO_4$, AR | 0.04 | -4.3 |
| Mili-Q water +Acid-2 | - | 100 uL 85 % $H_3PO_4$, HPLC | 0.03 | -4.9 |
| Mili-Q water +OX+Acid-1 | 2.0 g $K_2S_2O_8$ | 100 uL 85 % $H_3PO_4$, AR | 0. 63 | -25.90 |
| Mili-Q water +OX+Acid-1 | 2.0 g $K_2S_2O_8$ | 100 uL 85 % $H_3PO_4$, AR | 0.54 | -25.69 |
| Mili-Q water +OX+Acid-1 | 2.0 g $K_2S_2O_8$ | 100 uL 85 % $H_3PO_4$, AR | 0.46 | -24.77 |
| Mili-Q water +OX+Acid-2 | 2.0 g $K_2S_2O_8$ | 100 uL 85 % $H_3PO_4$, HPLC | 0.63 | -26.66 |
| Mili-Q water +OX+Acid-2 | 2.0 g $K_2S_2O_8$ | 100 uL 85 % $H_3PO_4$, HPLC | 0.56 | -27.38 |
| Mili-Q water +OX+Acid-2 | 2.0 g $K_2S_2O_8$ | 100 uL 85 % $H_3PO_4$, HPLC | 0.58 | -26.91 |

[a, b] oxidant and acid are added to 50 mL Mili-Q water.

ND* : Not detected

Line 190: Mention again here again that KHP is a standard. It was introduced in the very beginning, but most people will not remember the abbreviation several pages later

**Response:** According to the reviewer's suggestion, we mentioned that KHP as a standard here. Please see line 240 in the revised MS.

The revised sentence:

Different concentrations of working standard (KHP) are tested to compare the flushing methods.

Line 203: "overheated" sounds like too high temperature, not too long heating times

**Response:** Thanks for the reminder, we replaced the saying of "overheated vials" with "vials heated for longer time". Please see line 252 in the revised MS.

The revised sentence:

Some caps of the sample vials are out of shape after heating for longer time (more than 60 min), suggests gas leak of the vials. High pressure can be built up in the headspace with the increase of the temperature during the long time heating, especially for the vials containing more carbon contents.

Paragraph 3.3/Figure 2: Looking at Figure 2, I cannot find the trends and effects described in the text. I think a statistical test would show no significant differences between concentration or d13C values at different heating times. I can also not detect that data are less scattered at 60min. Vs (10ug) is maybe more scattered at 120 min than for other heating times, but at the same time the isotope values at 120 min are much more stable. Vs - 30ug has some big outliers for 120, 90 and 30min, but no corresponding outliers in d13C. On the other hand, d13C (4ug) has outliers at 60 and 30 min, when Vs-4ug is particularly stable. Overall I see data point with comparable scatter and occasional outliers, I think the text over-interprets the results.

**Response:** We have to apologized that the Fig 2 was not drawn in a suitable range of y axis which resulted in the loosing information of the dramatically high delta values corresponding with the low carbon content. The revised graph was given below and the it could be seen in line 918 in the revised MS.

[Figure]

Figure 2. Carbon contents (a) and isotopic ratios (b) of KHP after different heating time.

The results of a statistical test are given below:

Table S1. Statistical results of isotope compositions after different heating time.

| Heating time | Carbon content | Average $\delta^{13}C$ | SD (n=5) | Total SD |
|---|---|---|---|---|
| 15 | 0 | -32.01 | | |
| | 4 | -28.86 | 0.24 | |
| | 10 | -29.77 | 0.13 | 0.53 |
| | 30 | -30.00 | 0.02 | |
| 30 | 0 | -32.50 | | |
| | 4 | -28.92 | 0.53 | |
| | 10 | -29.71 | 0.09 | 0.57 |
| | 30 | -30.04 | 0.08 | |
| 60 | 0 | -31.03 | | |
| | 4 | -29.04 | 0.59 | |
| | 10 | -29.71 | 0.10 | 0.51 |
| | 30 | -29.96 | 0.05 | |
| 90 | 0 | -32.47 | | |
| | 4 | -28.77 | 0.22 | |
| | 10 | -29.85 | 0.28 | 0.71 |
| | 30 | -29.39 | 0.94 | |
| 120 | 0 | -29.13 | | |
| | 4 | -28.59 | 0.21 | |
| | 4 | -26.42 | 6.46 | 4.58 |
| | 30 | -27.30 | 5.34 | |

According to this table, the isotopic results after 60 min heating time were the most stable compared with other heating times. And that was also part of the reason why we chose to heat the samples for 60 min to oxidize the organic components in the optimized method.

Besides, 4ug C was lower than the detection limit of this optimized method, the isotope results were easier to be affected by the procedural blank. Thus the isotope results of 4 ug-standards were not as stable as the standards with larger amount of carbon content.

Line 221: Confusing formulation – the carbon content of the WSOC samples stays the same. Only the contamination detected during flushing is lower for samples stored less than 12 hrs . . .

**Response:** This sentence was restructured as "The carbon contents produced in the mixtures that stored less than 12 h before analysis is smaller than 0.02 µg, which contributes to ~ 7% to the carbon content in the procedural blank (0.5µg C).". Please see line 283-385 in the revised MS.

Line 225: purged? Do you mean "heated and analyzed"?

**Response:** Actually, we were talking about the waiting time before helium flushing step here (duration time between step 4 and step 5 in Fig. 1.). To avoid the misunderstanding, we changed the use of "purge" with "be flushed with He within 12 h …" here. Please see line 289 in the revised MS.

The revised sentence:

Therefore, the mixture should be flushed with He within 12 h to avoid $CO_2$ loss and isotope fractionation.

[Figure]

**Figure 1.** Schematic of the optimized method for the measurement of WSOC mass concentrations and the $\delta^{13}C_{-WSOC}$ values.

Line 226/227: "tested through reference gas detection" What do you mean by this?

**Response:** "detection" was deleted here, what we want to express was the setting of the loading times were tested with reference gas (i.e., $CO_2$ mixed with high purity helium).

Line 227/228: The sensitivity of a mass spectrometer does not depend on sample peaks and loading time. What do you mean by "sensitivity of mass spectrometry"?

**Response:** Sorry for the incorrect use of "sensitivity of mass spectrometry". What we want to illustrate here was the carbon contents measured with shorter loading time (e.g. 30-90s) were about 2 µg C lower than the 120 loading time. And the "sensitivity of mass spectrometry" was replaced by the description of the results with lower peak areas / carbon contents. Please see line 292-293 in the revised MS.

The revised sentence:

However, the amount of $CO_2$ in the reference gas detected by the mass spectrometry is about 2 µg C lower compared with longer loading times and more sample peaks.

Line 228/229: what do you mean by "decline in isotopic ratios"? A decrease in delta values (i.e. depletion)? Or a decrease in reproducibility? Or something else?

**Response:** "decline in isotopic ratios" means "a decrease in delta values" here. In case of misunderstanding, we restructure the sentence as "And there is a decrease of isotope results …". Please see line 294 in the revised MS.

The revised sentence:

And there is a decrease of isotope value ($\sim 0.4$ ‰) as well when the loading time is shorter or the samples peaks are less than 10.

Section 3.5.1: This section describes the method for blank correction, but barely any results. Please also give the results at least in the supporting material

**Response:** The isotopic ratio (-27.43‰) and the carbon content (0.3-0.5 µg C) of the blank were given in the section 3.5.2 (line 347) in the revised MS.

We added "For example, $\delta^{13}C_{blk}$ and $A_{blk}$ are calculated to be -27.43‰ and 0.3Vs (~0.5 µg C) based on the results of KHP and $CH_6$ (shown in Fig. 3.)."

Line 248 – 251: Show a graph of d13C_meas vs 1/A in the appendix, what was the typical R2?

**Response:** According to the reviewer's suggestion, a graph showing the correlation of isotopic ratios and 1/A was added in the appendix, the typical $R^2$ was 0.97. Please see S3 in the revised supporting information.

[Figure]

Figure S3. Relationship between the values of $1/A_m$ and $\delta^{13}C$ obtained from the measurement of $CH_6$.

Line 255-256: What were the values you obtained for d13C_blk and A_blk? How does A_blk compare to A of a typical sample?

**Response:** The carbon content and the isotope composition of the blank varied every day (though it was not varied strongly), so we added the $\delta^{13}C_{-blk}$ (-27.43‰ ) and A-blk (0.3 Vs,corresponding with 0.3-0.5 μg C) values obtained from the measurement of the standards as an example in section 3.5.2 and section 3.6. The peak area of a typical sample measured in this optimized method was larger than 3 Vs, which was 10 fold of the procedural blank. Please see section 3.5.2 (line 347) in the revised MS.

We added "For example, $\delta^{13}C_{blk}$ and $A_{blk}$ are calculated to be -27.43‰ and 0.3Vs (~0.5 μg C) based on the results of KHP and $CH_6$ (shown in Fig. 3.).".

Section 3.5.2: This section is very superficially described and very difficult to understand, partially due to poor English. I'm not 100% sure what the authors are doing here. It definitely needs to be carefully rewritten with more precise language and more detail. But if I am not completely mistaken by the vague descriptions, this section just describes a calibration, where delta values are normalized against standard material that is treated the same way as the samples?

**Response:** Sorry for the confusion due to the unclear expression. In this section, we described a calibration of unknown samples' isotopic ratios. This calibration was based on the calibration curve established by the isotopic ratios of different standards (correct values from EA-IRMS and measured values from GB-IRMS). According to the reviewer's suggestion, the section was rewritten and added with more detailed information. Please see section 3.5.3 (line 352-381) in the revised MS.

The revised Section:

3.5.3 Calibration of isotope results

In order to calibrate the isotope results, four working standards (KHP, BA, $CH_6$ and $C_2$) containing different carbon contents are measured with EA-IRMS and Gas Bench II-IRMS. The standards measured with EA are combusted at 1000℃to convert the organic materials into $CO_2$ for the measurement in IRMS without pretreatment. More than 10 repetitions of each standard are measured in this way, the average delta values (after blank correction) of each standard are defined as correct values here. On the other hand, the average isotope compositions (after blank correction) of 10 repetitions obtained from the wet oxidation method (determined with Gas Bench II) are defined as measured values. Thus the calibration curve can be established on the basis of the measured values and the correct values (Fig. S4.). For instance, the isotope results can be calibrated as follows:

$$\delta^{13}C_{cali} = k \times \delta^{13}C_{blk-corr} + b \qquad (8)$$

$\delta^{13}C_{cali}$ is the isotope composition after the isotope calibration, $\delta^{13}C_{blk-corr}$ is the blank corrected isotope composition determined with Gas Bench II, k and b are the slope and the intercept obtained from the calibration curve. Similar with the blank correction, the isotope calibration curve needs to be established with each batch of the ambient samples to assure the stable status of the IRMS and the proper processes during the pretreatment.

In this way, the isotope results can be calibrated, the raw data and the isotope composition after the blank correction and the isotope calibration determined with Gas Bench II are compared in Fig. 3. The correct values of standard carbon isotopes are plotted in Fig. 3. as well. The isotope results after two steps of correction (the blank correction and the calibration of isotope results) are closer to the correct values (isotopes measured with EA) and the blank contribution are drastically eliminated. But as for the standards containing carbon content smaller than 5 μg C, the contribution of the procedural blank (with an isotope ratio about -27.43‰) is still significant. According to the isotope variation of the ambient aerosols, the analysis of isotope compositions is not reliable if the repetitions of the standards show difference larger than 1‰ (SD > 0.5 ‰). After correction, the standard deviations of isotope results of each standard are better than 0.17 ‰ (regardless of the carbon content of a certain standard) when the carbon contents are larger than 5 μg C. In that case, the detection limit of this method is 5 μg C and the results (both carbon contents and the isotopic ratios) of WSOC lower than 5 μg C were not reliable.)

Line 259: I don't think "system errors" is the correct term to describe what you are investigating here, "systematic bias" is probably a better term.

**Response:** Thanks for the reviewer's suggestion, we have changed the title to "Calibration of isotope results" since this entire section was talking about establishing a calibration curve to calibrate the isotope results obtained from the Bas Bench II-IRMS. Please see line 352 in the revised MS.

The revised title:

**3.5.3 Calibration of isotope results**

Line 261/262: The logic here is reversed: Systematic errors are the cause of differences between measurements on different systems, not the other way around.

**Response:** This sentence was deleted in the revised MS.

Line 264: I assume larger amounts of the standard material analyzed on the EA (presumably without pretreatment) are taken as the "correct value" against which the standard materials that have undergone the whole extraction procedure are evaluated. If this is correct, then please describe clearly.

**Response:** Sorry for the confusion due to the unclear expression. We actually measured more than 10 repetitions of each standard on the EA-IRMS (without pretreatment except wrapping the solid standard in tin capsules) to determine the "correct value" of the specific standard. And on the other hand, we also measured the isotope compositions of all the standards with the optimized method introduced in this paper. We have rewritten this whole paragraph and described this part more clearly in the revised MS in line 353-362.

The revised paragraph:

In order to calibrate the isotope results, four working standards (KHP, BA, $CH_6$ and $C_2$) containing different carbon contents are measured with EA-IRMS and Gas Bench II-IRMS. The standards measured with EA are combusted at 1000℃ to convert the organic materials into $CO_2$ for the measurement in IRMS without pretreatment. More than 10 repetitions of each standard are measured in this way, the average delta values (after blank correction) of each standard are defined as correct values here. On the other hand, the average isotope compositions (after blank correction) of 10 repetitions obtained from the wet oxidation method (determined with Gas Bench II) are defined as measured values.

Line 266: what do you mean by "isotope standard curve", a calibration curve? Please show an example, at least in the appendix.

**Response:** "Isotope standard curve" here referred to a calibration curve to calibrate the isotope results. To avoid the misunderstanding, we revised it as "calibration curve" and showed an example in the supplement (S4). Please see line 360-366 in the revised MS and the supporting information.

The revised sentences:

Thus the calibration curve can be established on the basis of the measured values and the correct values (Fig. S4.). For instance, the isotope results can be calibrated as follows:

$$\delta^{13}C_{cali} = k \times \delta^{13}C_{blk-corr} + b \tag{8}$$

$\delta^{13}C_{cali}$ is the isotope composition after the isotope calibration, $\delta^{13}C_{blk-corr}$ is the blank corrected isotope composition determined with Gas Bench II, k and b are the slope and the intercept obtained from the calibration curve.

[Figure]

Figure S4. Calibration curve of the isotope composition for the correction of the systematic bias.

(The values of y axis are the correct isotope values of the standards measured with EA using combustion method without pretreatment, and the values of x axis are the measured isotope results of the standards obtained from the wet oxidation method and determined with Gas Bench II.)

Line 267: Which "two different peripherals"? This whole sentence does not make sense to me.

**Response:** "Two different peripherals" here were the EA-IRMS and the Gas Bench II-IRMS, and this sentence was restructured. Please see line 371-374 in the MS.

The revised sentences: The isotope results after two steps of correction (the blank correction and the calibration of isotope results) are closer to the correct values (isotopes measured with EA) and the blank contribution are drastically eliminated.

Line 269: I don't see raw values from EA in the figure, does the EA result not give the nominal value (horizontal line)? "Corrected results . . ." In figure 3 the corrected data are labeled as: "blank corrected" but the text suggests that they are both blank corrected and normalized (calibrated). Which one is true?

**Response:** Sorry for not writing this part clearly, raw data from EA-IRMS were not shown in Fig. 3. The values labeled as "blank corrected" were the values after two steps of correction (i.e. blank correction and isotope calibration). We have rewritten the text describing the Fig. 3., and have corrected the labels. Please see line 369-371 and 921 in the revised MS.

The revised sentences:

In this way, the isotope results can be calibrated, the raw data and the isotope composition after the blank correction and the isotope calibration determined with Gas Bench II are compared in Fig. 3. The correct values of standard carbon isotopes are plotted in Fig. 3. as well.

[Figure]

**Figure 3.** Isotope results before and after the two-step correction of the four standards.

(a. KHP, b. BA, c. CH$_6$, d. C$_2$. Red circle with a spot represents the two-step corrected isotopic ratios; ■, ◆, △, × represent the raw data from Gas Bench II; the dotted line represents the blank corrected $\delta^{13}C$ values tested by EA)

Line 272: What do you mean by "dilution curves". Dilution curves do not have a precision . . .

**Response:** Sorry for the use of ambiguous terms, what we wanted to express here was, the isotope results of the standards (containing different carbon content) have a relatively low standard deviation of 0.17‰. We have restructured the sentence, please see line 377-380 in the revised MS.

The revised sentences: After correction, the standard deviations of isotope results of each standard are better than 0.17 ‰ (regardless of the carbon content of a certain standard) when the carbon contents are larger than 5 μg C.

Section 3.6 This section contains a summary of the previous test results. What this section should contain is a description of how the quality of the unknown samples are assured. How is the blank correction done exactly, i.e. which values are taken for d13C_blank and A_blank? How many and which working standards are measured with the samples, and how frequently i.e. was the calibration curve described above only established once, or is it measured every day? Etc . . . please re-write this section.

**Response:** Following the reviewer's suggestion, section 3.6 was rewritten, the method of assuring the carbon content of unknown samples, the blank correction, the calibration curve and other detailed information were included in the revised MS.

The carbon content in unknown samples were quantified by the standard curve (Peak areas Vs Carbon content in the standard solution). And this standard curve was given in the supporting information (S2).

The blank contribution was corrected as described in section 3.5.2, the $\delta^{13}C_{-blk}$ and $A_{blk}$ was calculated to be -27.43‰ and 0.3 Vs. Please see line 347 in the revised MS.

The revised section 3.6:

3.6 QA/QC procedure

A batch of working standards with different carbon contents are measured to evaluate the optimized method in this study (data shown in Fig. 3.). The quality of the unknown samples is assured with a standard curve established with the peak areas and the corresponding input carbon contents of WSOC extract (e.g. in Fig. S2.). The conversion efficiency is 104 ± 3% during the oxidation of WSOC extract. The average recovery of the working standards and the ambient samples are tested to be 97 ± 6 % and 99 ± 10 %, respectively. The conversion efficiency and the recoveries suggest completely oxidation of WSOC extract without significant isotope fractionation in the pretreatment. The blank contribution is evaluated with the peak area and the isotopic ratio as calculated with the indirect method introduced in Sect. 3.5.2. According to the carbon content (0.3 - 0.5 µg C) and the isotope composition (~ -27.43 ‰) of the procedural blank, the WSOC detection limit of this method is 5 µgC, 10 times of the carbon content in the procedural blank. The blank corrected isotope compositions are calibrated again with the calibration curve as described in Sect. 3.5.3.

In order to obtain the carbon contents and the corrected isotope compositions of the unknown samples, at least two kinds of standards need to be measured before every batch of the unknown samples. The range of the carbon contents and the isotope compositions of the standards are required to cover the range of WSOC and $\delta^{13}C_{\text{-WSOC}}$ in the ambient samples, e.g. KHP, BA and $CH_6$. Hence, the concentration standard curve, the linear equations for the blank correction and the isotope calibration curve are able to be established according to the results of the standards. Besides, one standard should be measured after every 10 samples to assure the stable status of the equipment.

As for the isotope measurement, the precision of the last eight sample peaks is < 0.15 ‰ within a run for standards containing more than 1 µg C; between runs, the deviation of the standards with different carbon contents (> 5 µg C, n ≥ 10) is < 0.17 ‰. The accuracy is estimated to be better than 0.5 ‰ by comparing the calibrated $\delta^{13}C$ results from Gas Bench II and EA. Isotope results tested by Gas Bench II is slightly lower compared to the results of EA. A couple aerosol filters are tested to evaluate the reproducibility of the ambient samples as well. The reproducibility of the WSOC concentration and the isotope results of the ambient samples are 0.25 ± 0.04 µg C and 0.14 ± 0.07 ‰ respectively. To conclude, the presented method is considered to be precise and accurate to detect the low abundance of WSOC as well as isotopes in aerosol samples.

To test the applicability of this method measuring the atmospheric WSOC, the ambient aerosol samples collected in Nanjing are analyzed. And the WSOC concentrations are measured with TOC analyzer (Shimadzu) for comparison. Figure 4. shows the scattered plot of WSOC concentrations measured with the two peripherals (TOC analyzer and Gas Bench II-IRMS). The strong correlation ($R^2$ = 0.95, p<0.01) and the slope (0.97) demonstrate the reliability of measuring WSOC with the presented method. It suggests complete oxidation of

WSOC in aerosol samples, which means no significant carbon isotope fractionation happens during the preparation. Moreover, the $\delta^{13}C_{-WSOC}$ values (between -26.24 ‰ to -23.35 ‰) of ambient aerosols are close to the published data (from -26.5 ‰ to -17.5 ‰) (Kirillova et al., 2013; Kirillova et al., 2014). In that case, the $\delta^{13}C$ values resulted from this method are considered to be effective for ambient WSOC.

[Figure]

Figure S2. Standard curve to quantify the unknown samples.
(The standard curve is established by the $CO_2$ gas / the KHP solution and the input carbon content of the certain vials. The blue dotted line is the linear fit of the results of $CO_2$ gas, and the black dotted line is the linear fit of the results of KHP solution.)

Line 349: At least for PM2.5, not for PM10

**Response:** We agree with the reviewer, and revised the sentence as "CC is negligible to the amount of TC in $PM_{2.5}$". Please see line 481 in the revised MS.

Line 355: What could such a source be?

**Response:** It could be the WIOC emitted from vegetation. Please see line 488 in the revised MS.

We added: In that case, the $^{13}C$ depleted source which only contributes to non-WSOC components, such as WIOC emitted from the vegetation, is likely to be another reason of $\delta^{13}C_{-TC}$ depletion during the sampling period.

Line 376: primary OC from coal combustion does not contain much WSOC

**Response:** We agree with the reviewer, primary OC from coal combustion does not contain much WSOC. But the "coal combustion" was mentioned here as a possible source (regardless of primary source or secondary source) of the ambient WSOC instead of a major primary source.

**Minor comments:**

Line 84: "Previous method*s*". Please have the manuscript proofread for similar grammar errors throughout

**Response:** Thanks for your kind reminder, we have checked and corrected the grammar errors like this in the full text.

Line 162: "optimistic" – do you mean "optimal"?

**Response:** Yes, and we have changed the word as "optimal" in line 162 (of the original MS) and other places where we mentioned the optimal conditions. Please see line 200 in the revised MS.

The revised sentence:

Several tests are performed to adjust the optimal conditions for measuring high time-resolved WSOC aerosols with relative low carbon amounts.

Line 234: replace "blank effects of the" with "blank contribution to"

**Response:** Following the reviewer's suggestion, we have revised "blank effects of the" with "blank contribution to". Please see lines 319-320 in the revised MS.

The revised sentence:

The blank contribution to the WSOC mass concentrations and the $\delta^{13}C_{\text{-WSOC}}$ values are evaluated with the peak area and the isotope value of the procedural blank.

Line 235: " ..could represent" -> is proportional to

**Response:** Following the reviewer's suggestion, we changed the expression from "could represent" to "is proportional to" and enriched the description of evaluating the carbon contents. Please see lines 320 in the revised MS.

The revised sentence:

The peak area (average value of the last eight peaks) from the measurement is proportional to the carbon content in the vial and then is taken to represent the $CO_2$ amounts in the inflow of IRMS.

Line 411: "Exclusively" -> clearly

**Response:** Following the reviewer's suggestion, we changed the "Exclusively" to "clearly" in line 545 and other places describing this special period.

[revised manuscript text omitted]
 because of the low carbon contents in the vials, which would produce less $CO_2$ and therefore lower pressure in the headspace after long time of heating. Accordingly, heating time longer than 60 min increases the probability of gas leak in the measurement.

In the aspect of the isotope composition, KHP standards heated for 15min, 30min and 60 min all show stable results with similar standard deviations (from 0.51 - 0.57, see Table 1 in supplement). While, the heating time of 15min and 30 min are not long enough for the complete oxidation, which is shown in lower carbon content (Fig. 2.). Therefore, heating for 60 min at 100℃

is found to be the most suitable to produce constant results without isotope fractionation.

3.4 Waiting time and instrument settings

The waiting time of the mixture (aerosol extract and the oxidizing solution) between step 4

and 5 in Fig. 1. is tested to prevent $CO_2$ loss during the flushing. Some of the compounds in aerosol samples could be oxidized at room temperature. The $CO_2$ generated from the mixture before heating could be lost during the flushing step (Sharp, 1973). The ambient sample is tested to detect the room - temperature - oxidized $CO_2$ (Fig. S1.). Replicates of ambient aerosol extract from one filter are mixed with the oxidizing solution, and the mixtures of the aerosol extract and the oxidizing solution are flushed with He to exclude the effect of $CO_2$ (both in the headspace and in the mixture) as soon as possible. After flushing, the mixtures are stored at room temperature from 1 to 31 hours before analysis without heating. The carbon contents produced in the mixtures that stored less than 12 h before analysis is smaller than 0.02 μg, which contributes to ~ 7% to the carbon content in the procedural blank (0.5μg C ). But when the waiting time is extended to 31 h, up to 2.3 μg C (about 5 times of the procedural blank) is oxidized into $CO_2$. The room - temperature - oxidized $CO_2$ produced during the waiting time would be flushed out by the He in the later procedure and then would result in significant isotope fractionation in the delta results.

Therefore, the mixture should be flushed with He within 12 h to avoid $CO_2$ loss and isotope fractionation.

In addition, various combinations of shorter loading times (30-90 s) and/or fewer sample peaks (i.e. 5 sample peaks) are tested with reference gas ($CO_2$ mixed with He) to shorten the analysis in the system. However, the amount of $CO_2$ in the reference gas detected by the mass spectrometry is about 2 μg C lower compared with longer loading times and more sample peaks.

And there is a decrease of isotope value (~ 0.4 ‰) as well when the loading time is shorter or the samples peaks are less than 10. Thus, 120 s loading time and 10 sample peaks are necessary for the precise results, and the standard deviation is < 0.03 ‰ for the 10 sample peaks within a run.

3.5 Calibration of the results

3.5.1 Quantification of carbon content

The sample peak area is proportional to the carbon content in the vial and then is used to quantify the amount of $CO_2$ in the inflow of IRMS. The average value of the peak areas for the last eight sample peaks is taken as the peak area of a certain sample. The first two sample peaks are excluded to avoid the effect of the residual $CO_2$ of the former sample. We established a carbon content standard curve (linear equation) by measuring the peak areas of $CO_2$ gas samples containing 1-24 μg C (Fig. S2.). It has to be noted that the $CO_2$ gas samples containing larger carbon contents are not tested for the difficulty of injecting too much volume of $CO_2$/He gas. Then the amount of $CO_2$ oxidized from the unknown samples can be quantified with this linear equation (i.e., Carbon content (μg) = Peak area (Vs) × (2.50 ± 0.08) – (0.62 ± 0.39), $R^2$=0.98). The standard curve (linear equation) of the peak areas against the carbon contents in the WSOC solution (KHP

solution containing 1-100 μg C) is also established (Fig. S2.). And a linear equation similar with the peak areas against $CO_2$ gas is obtained (i.e., Carbon content (μg) = Peak area (Vs) × (2.34±

0.01) – (0.86± 0.14), $R^2$=1.00).

Then the conversion efficiency of the WSOC oxidation can be roughly calculated. The average conversion efficiency of WSOC solutions containing 1-100 μg C is 104 ± 3 %. 
[revised manuscript text omitted]

**7 Pages**

**1 Table**

**5 Figures**

**Table S1**. Statistical results of isotope compositions after different heating time.

| Heating time | Carbon content | Average $\delta^{13}C$ | SD (n=5) | Total SD |
|---|---|---|---|---|
| 15 | 0 | -32.01 | | 0.53 |
| | 4 | -28.86 | 0.24 | |
| | 10 | -29.77 | 0.13 | |
| | 30 | -30.00 | 0.02 | |
| 30 | 0 | -32.50 | | 0.57 |
| | 4 | -28.92 | 0.53 | |
| | 10 | -29.71 | 0.09 | |
| | 30 | -30.04 | 0.08 | |
| 60 | 0 | -31.03 | | 0.51 |
| | 4 | -29.04 | 0.59 | |
| | 10 | -29.71 | 0.10 | |
| | 30 | -29.96 | 0.05 | |
| 90 | 0 | -32.47 | | 0.71 |
| | 4 | -28.77 | 0.22 | |
| | 10 | -29.85 | 0.28 | |
| | 30 | -29.39 | 0.94 | |
| 120 | 0 | -29.13 | | 4.58 |
| | 4 | -28.59 | 0.21 | |
| | 4 | -26.42 | 6.46 | |
| | 30 | -27.30 | 5.34 | |

[Figure]

**Figure S1.** Carbon contents of one ambient aerosol sample replicates tested after different waiting time (duration between the mixture of aerosol extractions with the oxidizing solution and the helium flushing step) without heating.

[Figure]

Figure S2. Standard curve to quantify the unknown samples.

(The standard curve is established by the $CO_2$ gas / the KHP solution and the input carbon content of the certain vials. The blue dotted line is the linear fit of the results of $CO_2$ gas, and the black dotted line is the linear fit of the results of KHP solution.)

[Figure]

**Figure S3**. Relationship between the values of $1/A_m$ and $\delta^{13}C$ obtained from the measurement of
$CH_6$.

[Figure]

**Figure S4**. Calibration curve of the isotope composition for the correction of the systematic bias.

(The values of y axis are the correct isotope values of the standards measured with EA using combustion method without pretreatment, and the values of x axis are the measured isotope results of the standards obtained from the wet oxidation method and determined with Gas Bench II.)

[Figure]

**Figure S5**. Time series of WSOC/OC.